*EGU Atmospheric Measurement Techniques: 2022 – 05– 02. First revised version.*

# Detection of supercooled liquid water containing clouds with ceilometers: Development and evaluation of deterministic and data-driven retrievals

Adrien Guyot[1], Alain Protat[1], Simon P. Alexander[2], Andrew R. Klekociuk[2], Peter Kuma[3], Adrian McDonald[4]

[1] Australian Bureau of Meteorology, Melbourne, Victoria, Australia

[2] Australian Antarctic Division, Kingston, Tasmania, Australia

[3] Department of Meteorology, Stockholm University, Stockholm, Sweden

[4] University of Canterbury, Christchurch, New Zealand

*Correspondence to*: Adrien Guyot (adrien.guyot@bom.gov.au)

**Abstract.** Cloud and aerosol lidars measuring backscatter and depolarization ratio are most suitable lidars to detect cloud phase (liquid, ice, or mixed phase). However, such instruments are not widely deployed as part of operational networks. In this study,
we propose a new algorithm to detect supercooled liquid water containing clouds (SLCC) based on ceilometers measuring only co-polarisation backscatter. We utilise observations collected at Davis, Antarctica, where low-level, mixed phase clouds, including supercooled liquid water (SLW) droplets and ice crystals remain poorly understood, due to the paucity of ground-based observations. A 3-month set of observations were collected during the austral summer of November 2018 – February 2019, with a variety of instruments including a depolarization lidar and a W-Band cloud radar which were used to build a 2-
dimensional cloud phase mask distinguishing SLW and mixed phase clouds. This cloud phase mask is used as the reference to develop a new algorithm based on the observations of a single polarisation ceilometer operating in the vicinity for the same period. Deterministic and data-driven retrieval approaches were evaluated: an extreme gradient boosting (XGBoost) framework ingesting backscatter average characteristics was the most effective method at reproducing the classification obtained with the combined radar-lidar approach with an accuracy as high as 0.91. This study provides a new SLCC retrieval
approach based on ceilometer data and highlights the considerable benefits of these instruments to provide intelligence on cloud phase in polar regions that usually suffer from a paucity of observations. Finally, the two algorithms were applied to a

full year of ceilometer observations to retrieve cloud phase and frequency of occurrences of SLCC: SLCC was present $29 \pm 6$ % of the time for T19 and of $24 \pm 5$ % of the time for G22-Davis over that annual cycle.

1 Introduction

Mixed-phase clouds play a critical role in the earth radiation budget, through their complex interactions with incoming and outgoing shortwave and longwave radiation. This effect is particularly important at higher latitudes with variation in radiation affecting the snow or ice mass balance in the polar regions (Lawson and Gettelman, 2014). Despite their importance in the
global climate system, the occurrence, amount, and nature of mixed-phase clouds remain poorly simulated in global climate models due to the paucity of reliable mixed-phase clouds observations, especially in remote regions of the globe such as Antarctica (Bodas-Salcedo et al., 2016; Hyder et al., 2018). Until recently, global climate models assumed that low-level clouds over the Antarctic Ice sheet essentially contained ice crystals, but Lawson and Gettelman (2014), and later Ricaud et al. (2020) both showed from their observations that around 50% of clouds contained supercooled liquid water (SLW) during
the austral summer. Satellite-based lidar observations of mixed-phase clouds suffer from severe biases (Hu et al. 2009; Mace et al., 2020; Mace et al., 2021; McErlich et al, 2021) but points towards a high-frequency occurrence of mixed-phase clouds in Southern latitudes. Lawson and Gettelman (2014) and Ricaud et al. (2020) highlighted the significant impact the increased proportion of SLW clouds had on climate model simulations. The parameterization of cloud microphysical processes and precipitation remains challenging in Antarctica, given the limited observations; recent work (Sotiropoulou et al., 2021; Vignon
et al., 2021) focused on improving the parameterisation of SLW, showing how simulations and observations can be combined to improve our understanding of underlying processes leading to its formation. Kay et al. (2016) and Frey et al. (2018) also highlighted the importance of Southern Ocean mixed-phase clouds in global coupled climate models, under the predicted increase of greenhouse gases concentrations (Bjordal et al., 2021).

Depolarisation lidar is the most reliable means of observing non-spherical shape for randomly oriented cloud particles (Mishchenko et al., 2000; Hu et al. 2009; Mace et al. 2020). Typically, a depolarization ratio below 10% is characteristic of SLW clouds (Ricaud et al., 2020), while higher values are produced by ice particles. However, the remoteness and year-round harsh conditions for operating ground or aircraft operations have limited the frequency of cloud observation campaigns in Antarctica. Only over the last decade have coordinated ground-based cloud and precipitation studies been conducted in various
regions of Antarctica, including in Adelie Land (Grazioli et al., 2017; Genthon et al., 2018), Dronning Maud Land (Gorodetskaya et al., 2015), Ross Island (Scott and Lubin, 2014; Zhang et al., 2019), the South Pole (Lawson and Gettelman, 2014), the Antarctic Peninsula and Larsen Ice Shelf (Grosvenor et al., 2012; Lachlan-Cope et al., 2016) and East Antarctica (Alexander et al., 2021, Gehring et al., 2022). Complementary to ground-based observations, satellite-borne remote sensing

capabilities including depolarisation lidar and cloud radar can be combined to generate cloud phase products in remote regions such as the Arctic and the Antarctic, capitalising on polar orbit satellites revolutions, with high frequency flights over the poles (Litowski et al., 2019, 2020). Although these active remote sensing satellites enable generation of cloud phase products covering large areas, the drawbacks are reduced temporal and spatial resolutions as compared to ground-based remote sensing capabilities such as lidars and radars, and strong extinction of the lidar backscatter, making lower layers closer to the ground not observable from space if highly attenuating layers are present above. Satellite observations also suffer from the effect of ground clutter for observations closer to the ground (Bennartz et al., 2019). These recent cloud observations campaigns all pointed towards a higher than anticipated occurrence of SLW and mixed phase clouds (Scott and Lubin (2016); Ricaud et al., 2020; Zhang et al., 2019; Alexander et al., 2021; McErlich et al., 2021; Cossich et al., 2021).

Extensive observation campaigns require the deployment of dedicated instruments to determine the cloud phase, including usually, at least, a depolarization lidar. Hogan and Illingworth (1999) proposed to detect supercooled liquid water with ceilometers despite the absence of depolarisation data. Ceilometers are widely used and deployed by national weather services, typically at airports, to provide information on cloud cover and cloud base height. The manufacturers of ceilometers directly provide the cloud base height and cloud cover, using proprietary algorithms. These variables have been derived from the attenuated backscatter profile measured by the ceilometer. In an operational context, the attenuated backscatter profile is generally not used, even though it contains valuable information on the structure of the atmospheric boundary layer and the thermodynamics of the cloud phase (Hogan et al., 2003, 2004; Morille et al., 2007; Munkel et al., 2007; Van Tricht et al., 2014), as well as the presence of aerosols. Following the initial work from Hogan and Illingworth (1999), further studies led to the development and deployment of new detection algorithms for liquid cloud base layers and SLW (O'Connor et al., 2004), as part of the Cloudnet initiative (Illingworth et al., 2007). Recently, Tuononen et al. (2019) (thereafter referred to as T19) proposed an improved approach from the Cloudnet retrieval (Illingworth et al., 2007), utilising the shape of the attenuated backscattered profile instead of relying on finding the first value of backscatter above a given threshold value (Illingworth et al., 2007). Hamalainen et al. (2020) further applied the T19 approach combined with vertical meteorological profiles of temperature to build a hydrometeor classification scheme, to detect supercooled liquid water in the clouds. These recent studies were the starting point and the motivation for the present paper: to evaluate if the T19 approach could be successfully applied to ceilometer observations from Antarctica.

The aims of this study were to: (i) utilise high-resolution observations of cloud phase combining a set of ground-based instruments including a depolarization lidar and a cloud radar, to better understand cloud processes and microphysics; (ii) to evaluate the T19 supercooled liquid water retrieval for a ceilometer dataset collected at Davis, Antarctica; (iii) to develop, train, and test a new enhanced algorithm to retrieve supercooled liquid water, using only the attenuated backscattered signal

measured by a ceilometer and ERA5 interpolated temperature fields; (iv) to apply the cloud phase retrievals to one year of ceilometer observations and produce a climatology of SLCC for Davis.

## 2 Methods

### 2.1.  Data

#### 2.1.1 The PLATO data

As part of the Australian Antarctic Division's Precipitation over Land and The Southern Ocean (PLATO) field campaign, which operated during the Year of Polar Prediction (YOPP, Bromwich et al., 2020), a suite of ground-based remote sensing instruments was deployed at Davis (68.5762 ºS, 77.9696 ºE), one of the three permanent Australian Antarctic bases on the continent (Gehring et al., 2022), during the southern hemisphere summer 2018/2019. These included a W-Band radar (Delanoë et al. 2016), a 355 nm depolarisation lidar operating for 3 months, and a ceilometer operating during a full year from November 2018 to October 2019. This set of instrumentation provides a unique opportunity to understand the physics of clouds over that region, improve existing or develop new cloud phase retrieval algorithms, and test a variety of models. All these instruments are non-scanning devices, therefore only able to perform acquisitions along a single path.

During the early 2010s, the World Meteorological Organisation Weather and Research Program initiated a 10-year collaborative research project (2013-2022), the Polar Prediction Project (PPP), aiming at improving the prediction of weather for polar regions, at short (hourly) to longer (seasonal) timescales (Jung et al., 2016). The pinnacle of the PPP are intense observational, modelling, and other related activities conducted under the umbrella of the Year of Polar Prediction (YOPP). Three Special Observing Periods (SOPs) occurred between mid-2017 and mid-2019, and included deployment of dedicated instrumentation on the ground, together with higher frequency routine observations (Bromwich et al., 2020).

The cloud radar, namely the Bistatic Radar System for Atmospheric Studies (BASTA) was initially developed within a research laboratory and further became a semi-operational instrument (Delanoë et al., 2016). The radar has a sensitivity of around – 50 dBZ at 1 km, a vertical resolution of 25 m and a sampling frequency of 12 s. The BASTA was deployed on a dedicated concrete slab, and oriented to point vertically with an accuracy better than 0.1°, as shown in Figure 1b.

< Figure 1 here >

The Leosphere R-MAN510 cloud and aerosol depolarisation lidar (Royer et al., 2014) consists of elastic transmission and
reception (parallel and perpendicular) at 355 nm and inelastic (Raman) reception at 387 nm. The "raw" data from the RMAN
lidar is of variable temporal resolution generally ranging from about 30 to 90 seconds, and 15 m vertical resolution. The lidar
was operated with a 4-degree off zenith angle to avoid ambiguity between SLW clouds and oriented ice plates (Hogan and
Illingworth, 2003). We did not consider the Raman channel further in this analysis due to persistent daylight during its Davis
deployment rendering the weak Raman return signal unusable (Alexander and Protat, 2019). Raw lidar data were processed to
provide vertically resolved profiles of cloud phase, which broadly followed the algorithms developed by Alexander and Protat
(2018) and Noh et al. (2019). The raw 355nm lidar backscatter profiles were first processed to remove background noise and
correct for beam overlap. We calibrated the lidar following the method of O'Connor et al. (2004), who demonstrated that the
lidar ratio is constant within optically thick liquid non-precipitating stratocumulus clouds. We scaled the raw signal until the
observed lidar ratio matched the theoretical lidar ratio within these clouds. Then, the calibration values obtained during
stratocumulus for the limited number of optically thick clouds present above Davis were used to calibrate the three months of
data collected during the summer.

Following calibration, we used a speckle removal technique to flag spurious noise which is ubiquitous at high altitudes in both
the parallel and perpendicular channels. We followed the method of Alexander et al. (2021) who used a first pass of the
algorithm to extract bright clouds (with large vertical gradient in backscatter) in the co-polarised channel, and then assigned
cloud phase based upon the layer-averaged backscatter and depolarisation (Hu et al., 2010). We isolated additional
hydrometeors and aerosols based on pixels which had depolarisation ratios exceeding molecular backscatter and variances
within empirically determined thresholds. A region-of-interest analysis to extract conjoined regions removed any spurious
pixels initially flagged as hydrometeors or aerosols. The result of these steps was a much greater detection of ice virga than
only using the parallel backscatter because thin ice virga has large depolarisation ratios, making them readily detectable in the
perpendicular channel. Ice is defined to be virga rather than cloud when it exists beneath a SLW layer. It also allowed
attribution of liquid precipitation reaching the surface, because this second stage of the algorithm didn't require vertical
gradients of backscatter to determine the presence of hydrometeors.

### 2.1.2 Vaisala CL51 ceilometer observations

In operational settings, ceilometers usually report cloud base heights and oktas (percentage of cloud cover over a given area)
without providing information on cloud phase. However, these instruments record the full backscattered profile from which
the cloud base and okta have been derived. In this study, raw data was collected with the University of Canterbury Vaisala
CL51, e.g., the full backscattered profile with a range of 15 km, a vertical resolution of 10 m and a time resolution of 15 s.
These observations covered the PLATO period (November 2018 to February 2019) for which the depolarization lidar and the

W-Band radar operated and extended till October 2019. The data was pre-processed using a dedicated software developed by Kuma et al. (2021), namely the Automatic Lidar and Ceilometer Framework (ALCF). This software allows the processing of raw data from a variety of lidars and serves as a platform for comparing observations and models. Here, we used the version 1.1 of the software for processing the raw data generated by the Vaisala CL51 to: (1) produce daily netCDF files from the hourly Vaisala file format; (2) remove noise by applying a noise removal algorithm and subsampling the data to 5 min, 50 bins; The noise removal is done by estimating the distribution of noise at the highest available range and subtracting the mean

of the distribution (scaled by the square of range) from all bins in the column. In the cloud masking, the standard deviation of noise is considered when determining if a bin is cloudy. By default, five standard deviations are subtracted from the value before the cloud mask threshold is applied. This is done to prevent false positives with sufficient probability; Subsampling is mostly done to improve signal-to-noise ratio. The cloud masking usually benefits from subsampling to 5 min intervals and 50 m vertical resolution, because it decreases the number of misclassified bins; (3) After noise removal, then subsampling, ALCF

performs a calibration of the attenuated backscatter using the approach of Hopkin et al. (2019). In addition to the absolute calibration, the instrument built-in software applies overlap calibration internally. The final pre-processed products were daily netCDF files including the total attenuated volume backscattering coefficient (ß, $m^{-1}$ $sr^{-1}$) at a resolution of 5 min and bin vertical resolution of 50 m.

### 2.1.3 ECMWF ERA5

The latest-generation reanalysis product ERA5 from the European Centre for Medium-Range Weather Forecasts (ECMWF) was used in this study (Hersbach et al., 2020). The ERA5 hourly data on pressure levels were extracted via the Copernicus portal (https://cds.climate.copernicus.eu) as monthly netCDF files containing the geopotential, potential vorticity ($pv$, K $m^2$ $kg^{-1}$ $s^{-1}$), relative humidity ($r$, %), air temperature ($t$, K), the specific cloud ice water content ($ciwc$, kg $kg^{-1}$), the specific cloud liquid water content ($clwc$, kg $kg^{-1}$), the specific rain water content ($crwc$, kg $kg^{-1}$), the specific snow water content ($cswc$, kg

$kg^{-1}$), the horizontal components of the wind speed ($u$ and $v$, m $s^{-1}$) and the vertical velocity ($w$, Pa $s^{-1}$). During the YOPP, enhanced observations were conducted including four radio soundings per day at Davis, instead of two during normal periods. The YOPP covered approximately the period with a concomitant operation of the W-Band radar and depolarisation lidar.

The nearest ERA5 grid point (located at 68.5 ºS, 78.0 ºE) to the location of the ceilometer, W-Band radar and depolarisation

lidar was used as the centre of 9 neighbouring grid points forming a square. All the extracted variable fields were averaged to reduce potential spatial variability effects and reduce noise. A sensitivity to this averaging approach was also performed using only the central grid point and the averaging effect on the temperature and humidity fields was considered negligible. The vertical pressure level fields were linearly interpolated to the 50 m vertical resolution grid of the ceilometer, and the hourly variables were linearly interpolated to 5 min to match the ceilometer time resolution.

**2.2 Cloud phase masks**

**2.2.1. Radar-lidar merged cloud phase mask**

This cloud phase product is obtained by merging information obtained from the W-Band radar and the depolarisation lidar. The principle is the same as the approach from Delanoë and Hogan (2010) with satellite-based sensors, which combined
observations from CloudSat and the Cloud Aerosol Lidar and Infrared Pathfinder Satellite Observations (CALIPSO) mission lidar, taking advantages of the different sensitivities of the radar and the lidar. The underlying principle for SLW versus mixed-phase classification of a grid point is that the W-Band radar is not sensitive enough to detect very small, supercooled liquid water droplets. As a result, when a value of reflectivity is measured for the grid point labelled as SLW by the lidar, it implies that there must be ice particles in the volume generating a backscattered radar signal, mixed with SLW droplets as detected by
the lidar. In this paper, the pipeline to produce the cloud phase mask was based on the procedure described in Noh et al. (2019), with some modifications.

In the first stage, the lidar and radar are re-gridded to the same temporal and vertical resolution, to create a new merged grid at 15-m vertical resolution, and 1-min temporal resolution. The original 2 min resolution RMAN lidar classification was re-
sampled to 1 min simply by duplicating the nearest 2 min timesteps. ERA5 reanalysis data (hourly on pressure levels) were extracted for the closest grid point to Davis and linearly interpolated in time and space onto the merged radar-lidar grid. The second stage is to incorporate the cloud phase product from the R-MAN lidar, as described in detail in Section 2.1.1 above. In the third stage, the original lidar-only cloud phase labelling is refined utilising the cloud radar reflectivity field. If there is no measurement of reflectivity above the noise level for the grid point, we assume that there is no ice. In that case, points labelled
"SLW" in the lidar-only cloud phase classification remain labelled as "SLW". Conversely, the presence of an observed radar reflectivity implies that there is ice in the volume as well, which triggers a new classification of the grid point as "Mixed-phase". Finally, the last stage consists in utilising the radar reflectivity to identify signals at subfreezing temperatures, while the lidar backscatter is fully attenuated by lower clouds and doesn't provide any information on the cloud phase. This case triggers the grid point to be labelled "Unknown" as there is no possibility to distinguish Ice particles-only from a mixed-phase,
although there is certainty that these grid points are not containing only SLW (Noh et al., 2019). "Unknown" could therefore be interpreted as "Ice or mixed-phase" if needed.

It is necessary to determine the uncertainty in the Raman lidar liquid phase product before quantifying the performance of the ceilometer liquid cloud algorithm. To this end, we performed a Monte Carlo simulation using a random population of N =
1,000 samples from a normally distributed population (Alexander et al., 2021). For each integrated cloud attenuated backscatter $\beta_{int}$ and integrated depolarization ratio $\delta_{int}$ point, we have associated uncertainties $\Delta\beta_{int}$ and $\Delta\delta_{int}$, which we set to be twice

the standard deviation of the normal distribution. We then determined the cloud phase for each of these 1,000 realizations. These simulations indicated that we misclassified only around 1.3% of the SLW as ice with the Raman lidar during the three months of observations, showing the robustness of our radar-lidar cloud phase product.


### 2.2.2. Ceilometer cloud phase mask based on T19

The first cloud phase mask presented herein is based on ceilometer observations, following the work from T19, augmented with ECMWF ERA5 interpolated temperature fields to differentiate SLW from other liquid water. Liquid water droplets

generate very high values of the ceilometer backscatter signal, and subsequent strong attenuation in the vertical profile above the altitude of liquid water. T19 proposed a modification from the Cloudnet approach (Illingworth et al., 2007), utilising the shape of the attenuated backscatter profile, instead of using a single threshold value. The input to the technique is the pre-processed ceilometer dataset, e.g., the 50 m gate resolution, 5 min calibrated attenuated backscatter processed with ALCF as explained previously.


The exact approach proposed by T19 was implemented herein: the maximum of a localised peak value in the vertical profile of the backscatter is found, instead of the first value above a given threshold as in Cloudnet.  However, the maximum of the peak value needs to exceed the same threshold value as in Cloudnet, namely the pivot ß value of $ß = 2 \times 10^{-5}$ m$^{-1}$ sr$^{-1}$, together with a maximum peak width at half height set at 150 m.  The combination of these two criteria allows the identification of a

rapidly attenuating signal, which is typical of liquid water layers. In the case of T19, this method of identification enabled capture of the base of precipitating clouds, but the authors also noted its potential application for in-cloud icing detection. The authors also showed the possibility of identifying multiple peaks within the same profile with this method. However, they did not specify how to allocate a classification to the bins at the height of the peak and the surrounding bins (below and above the peak). We decided that based on the above, only the altitude bin corresponding to the location of the peak (if found) was

labelled as liquid water. Further to this, a reclassification was done to distinguish supercooled liquid water from other liquid water based on the interpolated re-analyses temperature fields: if the temperature T was between 0 ℃ and above – 38 ℃, the grid points were classified as supercooled liquid water, otherwise they were classified as liquid water.

In addition to liquid water, precipitation and ice clouds were identified following the same approach as T19, by selecting grid

points with values of backscatter above $ß = 3 \times 10^{-6}$ m$^{-1}$ sr$^{-1}$ with a thickness of at least 350 m, e.g. 7 consecutive grid points satisfying this criteria, therefore showing no attenuation within at least 350 m. The base of these clouds was accordingly the lowest grid point of the points within the profile satisfying these criteria. As noted by T19, liquid layers can be identified within precipitation and ice clouds as defined utilising our algorithm.

Fog is a phenomenon that probably occurs relatively frequently in the Southern Ocean and some regions of Antarctica (Lazzara, 2008), although few studies are available in the literature to accurately quantify its occurrence (Kuma et al., 2020). The same method as T19 was again used here, detecting fog layers by identifying values of backscatter above $\beta = 10^{-5}$ m$^{-1}$ sr$^{-1}$ for the lowest grid point (corresponding to 0-50 m above the surface) and a $\beta$ value 250 m above the instrument of $\beta < 3 \times 10^{-7}$ m$^{-1}$ sr$^{-1}$ (to restrict the identification to fog, and exclude low-level thicker clouds).


### 2.3 Enhanced data-driven ceilometer cloud phase mask (G22-Davis)

Based on detailed observations of the cloud phase mask from T19 for days with substantial amounts of clouds, a large amount of speckle in the retrieved SLW phase was observed, corresponding to timesteps for which the radar-lidar cloud mask did not observe any SLW. This led us to investigate if an alternative algorithm could perform better for these conditions. Importantly,

the concomitant high-resolution and robust observations of cloud phase using the combination of radar and lidar provided us with a reference that could be used to develop and validate our new algorithm.

This new algorithm relies on an initial signal analysis of each attenuated backscattered profile, as in T19, but also makes use of the statistical properties of the full dataset. It is based on a data-driven model including a learning and testing phase using

the reference radar-lidar cloud mask.

< Figure 2 here >

The first step is to build a dataset and design, train and test a supervised model. This first step is summarised in the flowchart

in Figure 2. First, we detected all peaks in the dataset that had at least a width of 50 m (thus only one range bin since this is the lowest resolution of our post-processed ceilometer data), and a peak value of $\beta = 2 \times 10^{-5}$ m$^{-1}$ sr$^{-1}$ (similar threshold value as in Tuononen et al. 2019). Several peaks can be detected for the same profile. For each peak, seven features were attributed to characterise peak properties: the value of the backscatter at peak location, the peak width, the value of the backscatter at peak width, the peak prominence (e.g. the difference between the peak value and the surrounding baseline), the peak altitude

above ground level, the total number of detected peaks for a given profile and if several peaks, the order of the peak within that total number, with the lowest peak in altitude taking the number '0'. The height at which the peak width is measured is relative to its prominence, following eq. (1):

$$H_{peak\ width} = H_{height} - Prom \times RH \tag{1}$$

With $H_{peak\ width}$ the height of the peak at which the width is measured (m$^{-1}$ sr$^{-1}$), $H_{peak}$ the absolute height of the peak, *Prom* is the prominence (m$^{-1}$ sr$^{-1}$), and *RH* the relative height, which was set at 0.5. In addition to this, the peak temperature was also extracted using interpolated ERA5 fields. This 8-feature dataset of peak properties (Table 1) was then labelled for each row with a Boolean attribute based on the detection of either supercooled liquid water or mixed phase by the radar-lidar mask for that timestep (True: detection of SLW/Mixed Phase, False: no detection of SLW/Mixed Phase).


Table 1: Features of peak properties, including full name, abbreviations, and units.

| Feature | Abbreviation | Units |
|---|---|---|
| Value of the backscatter at peak location | peak_value | m$^{-1}$ sr$^{-1}$ |
| Total number of peaks for a given time step | peak_total_number | unitless |
| Peak prominence | peak_prom | m$^{-1}$ sr$^{-1}$ |
| Peak width | peak_width | unitless (to convert to meters, multiply by 50 m) |
| Peak width height | peak_width_height | m$^{-1}$ sr$^{-1}$ |
| Peak elevation above ground level | peak_alt | m |
| Peak number (in ascending order starting from ground level) | peak_number | unitless |
| Peak temperature | Peak_t | degrees K |

We also accounted for the problem caused by signal extinction in multi-layer SLCC situations: peak properties of a single peak corresponding to SLCC with no extinction other than molecular in the lower levels, cannot be directly compared in terms of

backscatter values to peak properties of a presumed SLCC peak at higher altitude for which the signal would have undergone substantial extinction by clouds or the presence of SLW of mixed phase below. The properties of a peak that would have undergone attenuation will therefore see lower values of the value of the attenuated backscatter at peak location. To account for this attenuation effect, we compared the value of the backscatter at peak location for single peaks, and for peaks that would have experienced extinction (peaks with a peak order > 0). We call the first group of peaks that don't see lower-level attenuation

"primary peaks" and the group of peaks that are higher in altitude above primary peaks "secondary peaks". For primary peaks, data for which SLCC were identified were selected based on the Boolean condition defined using the radar-lidar cloud mask. For secondary peaks, an empirically based set of conditions must be defined to extract only potential SLCC peaks from the secondary peaks. These conditions were based on the observed statistical distribution of peak properties and were empirically set as: the width of the peak must be < 4 bins (corresponding to 200 m), the peak width height must be > 40 x 10$^{-6}$ m$^{-1}$ sr$^{-1}$, and

the peak prominence must be > 60 x 10$^{-6}$ m$^{-1}$ sr$^{-1}$. The secondary peaks group also include the cases of third and fourth peaks in elevation above the primary peaks when found. Third and fourth peaks were only found in 26 cases (third peak) and 5 cases (fourth peak), representing 0.6% and 0.1% of the cases from all identified peaks. This very small sample size did not allow to

use the approach proposed for second peaks and therefore 2nd, 3$^{rd}$ and 4$^{th}$ peaks were all included in the same "secondary

peak" group. The distributions of the two sets of data (primary and secondary peaks) and their Kernel Density Estimates

(KDEs) are shown in Figure 3.

< Figure 3 here >

For primary peaks, one would expect the distribution of values of the peak to vary based on the concentration of liquid water.

One would expect the same effect for secondary peaks, but the values at peak would be smaller due to varying degrees of

extinction of the backscatter signal at lower levels. The absolute difference between the median value of the primary peaks

distribution (Q2$_{prime}$) and the secondary peaks distribution (Q2$_{second}$) can be calculated as $\left| Q2_{prime} - Q2_{second} \right|$ and is equal

to $4.20 \times 10^{-5}$ m$^{-1}$ sr$^{-1}$ . Our hypothesis was that for such a large amount of data in both cases (primary and secondary peaks),

this difference, or offset, was the value of the average extinction due to the presence of various phases in the cloud on the

lower altitude that affects the potential SLCC peaks. Adding this offset to the value of the backscatter at peak location for all

datapoints of the secondary peaks' distribution would therefore "adjust" their peak values. This is shown as well in Figure 3;

with the added offset, the adjusted distribution of secondary peaks cover roughly the same area of the primary peak distribution.

While not perfect, this adjustment enabled us to modify our 8-feature dataset to allow a fairer comparison across datapoints

for the peak value, a critical feature for further analysis. Additionally, to this adjustment of the peak value for some of the

peaks, the secondary peaks for which the timesteps were labelled "true" which did not meet the arbitrary criteria also had their

labelling changed from "True" to "False". Following the above, a new "adjusted" peak properties dataset can be used for

further analysis.

The next step of our algorithm development was to design, train and test a data-driven model that could predict the label of

each of the peaks. A relatively novel tree-based ensemble method was proposed by Chen and Guestrin (2016), i.e. extreme

gradient boosting or XGBoost, which is an improved version of gradient boosting with the advantages of reducing overfitting

and computational costs. The excellent performances of this method for a wide range of applications, consistently

outperforming other methods such as Support Vector Machines or Random Forest led us to select this approach here. The

principle of this algorithm relies on a "boosting" strategy, where predictions of "weak" learners (here, "learners" are decision

trees) are combined to produce a "strong" learner by utilising additive training strategies. The computational cost is reduced

by allowing parallel computations during the training phase (Chen et al., 2015). Here, we only cover the fundamental principles

of the additive learning, and the reader should refer to Chen and Guestrin (2015) for more details. The first learner was initially

fitted to all input data; a second model was then fitted to the residuals to reduce the disadvantage of the "weak" learner. This

process of fitting was repeated several times until the model satisfied a predefined criterion. The prediction of the model for a given set of hyperparameters was obtained by combining the predictions of each learner. The function that describes the prediction at each step $t$ can be written as eq. (2):

$$f_i^{(t)} = \sum_{k=1}^{t} f_k(x_i) = f_i^{(t-1)} + f_t(x_i) \qquad (2)$$

where $f_t(x_i)$ is the learner at step $t$, $f_i^{(t-1)}$ and $f_i^{(t)}$ are the predictions at steps t-1 and t, and $x_i$ is the input variable.

The extreme gradient boosting model uses the below expression to evaluate the model performance, eq. (3):

$$Obj^{(t)} = \sum_{k=1}^{n} l(\overline{y_i}, y_i) + \sum_{k=1}^{t} \Omega(f_i) \qquad (3)$$

where $l$ is the loss function, $n$ is the number of observations and $\Omega$ is the regularisation term defined as eq. (4):

$$\Omega(f) = \gamma T + \frac{1}{2}\lambda||\omega||^2 \qquad (4)$$

where $\lambda$ is the regularisation parameter, $\gamma$ is the minimum loss needed to partition the leaf node and $\omega$ is the vector of the scores in the model leaves.

Features were passed to the model for training using a 3 Stratified k-folds cross validation. Stratified k-fold is a variation of k-fold (Ojala and Garriga, 2010) where each training/testing set (fold) contains approximately the same percentage of each target class as the complete set. The cross-validation k-fold approach allows us to train and test the model three times. In this approach, testing folds never share the same data with other folds. With the 3 k-fold cross validation, two third of the data are allocated to training, while the remaining one third of the data is used for testing. Given the low number of features and small size of the dataset, computational cost was not a limitation, therefore no principal component analysis was applied to the data prior to model training and testing, and an extended grid search over a wide range of hyperparameters was implemented utilising both accuracy and balanced accuracies as the scoring methods. The hyperparameters maximum depth, minimum child weight and learning rate (eta) were explored with respective chosen values of [0.3, 0.2, 0.1, 0.05, 0.01, 0.005] for the learning rate, [9,12] for the maximum depth and [5,8] for the minimum child weight. Other parameters were set at default values. Data preparation including splitting the data into training and testing for cross validation was implemented using the python library

scikit-learn (Pedregosa et al., 2011) and the XGBoost model was developed using its python library XGBoost (Chen et al., 2015). All simulations were performed using a 1.3 GHz dual core Intel Core M and 8 GB of RAM memory.

Finally, once our model has been trained and tested on the data, the third step was to apply the algorithm including the trained model subsequently to each profile. This approach is summarised in the flowchart in Figure 4.

< Figure 4 here >

The algorithm treated each vertical profile of attenuated backscatter sequentially: in the first step, peaks were detected, and
their associated properties computed. If no peaks were detected, the timestep corresponding to that backscatter profile was labelled as "not SLCC".  If several peaks were detected, a sequential pipeline as seen in Fig. 4 was implemented to correct potential SLCC peaks for extinction using the statistical properties obtained in the pre-processing stage. For that given timestep, one or more peaks could be identified together with their properties. These peak features were passed to the previously trained XGBoost model for labelling, either "SLCC" or "not SLCC". If a given peak was labelled as "SLCC", the
corresponding bin at peak altitude was labelled "SLCC", as well as the surrounding bins, with the SLCC lower and upper boundaries defined as twice the peak width (at mid height) value. With this algorithm, several layers of SLCC can be identified within a single profile, and SLCC layers can be identified within or outside a cloud. To facilitate the discussion in the next sections of this study, we further refer to our algorithm as G22-Davis. The extension "-Davis" illustrates that our G22 model had been trained with data collected at Davis, and we can imagine that the same model could be applied to data collected
elsewhere.

**2.4 Strategies for intercomparison of cloud masks**

As mentioned previously, the resolution of the merged radar-lidar mask was 1-min and 15-m, while the resolution of the ceilometer cloud mask was coarser (5-min, 50-m). For both masks, linearly interpolated hourly ERA5 variables fields have been used. In order to compare both masks various strategies regarding resolutions can be considered.

For the masks intercomparison, the coarser resolution of the ceilometer mask was used, and the radar-lidar mask was subsampled to 5-min timestamps. Since spatial variabilities could occur at the finer vertical resolution of the W-Band, depolarisation lidar and ceilometer, grid to grid comparison of the masks was not considered suitable and relevant given the objectives of this study. Instead, a comparison timestep to timestep was performed, integrating the information available over
each vertical column. Then, two different strategies were used to subsample the merged radar-lidar mask: (i) the matching timestamps of both masks were found, and if SLW or mixed-phase was identified in one bin of the merged radar-lidar mask,

that timestep was labelled as positive, otherwise it was labelled as negative. Similarly, if SLCC was identified in one of the bins of the vertical column for the ceilometer mask, that timestamp was labelled as positive; (ii) a condition on the spatio-temporal structure of SLW and mixed-phase bins was considered over 5-min periods: for the timestamp to be labelled as

positive, a given number of consecutive bins labelled SLW or mixed-phase need to be found at the same height. Threshold values for consecutive bins were set at 3, 4 and 5 and this criterion was applied as a 5-min moving window on the merged radar-lidar cloud mask to produce three subsets of data corresponding to this second strategy. Labelling of SLCC for the ceilometer cloud mask was performed similarly to the first strategy.

2.5 **Metrics to evaluate mask intercomparison and model performances**

Mask intercomparison, physical model evaluation and data-driven training and testing involve evaluating performance of predictions by comparing two one dimensional Boolean vectors (positive (true) for the presence of SLCC, negative (false) for the absence of SLCC). A few basic definitions are provided below, before the evaluation strategy, which is presented just after.


A true positive (TP) is defined as a test result indicating a correct prediction (correctly predicting the occurrence of SLCC), a true negative (TN) is defined as a test result correctly predicting the absence of SLCC, while a false positive (FP) is defined as a test result wrongly predicting the presence of SLCC, and a false negative (FN) is a test result wrongly indicating the absence of SLCC.


The precision is the ratio of the number of true positives over the number of true positives and false positives, i.e. the ability of the classification not to label as positive a negative sample and is defined as eq. (4):

$$precision = \ TP \ / \ (TP + FP)$$

(4)

The recall is the ratio of true positives over the number of true positives and false negatives, e.g. the ability of the classification to find all the positive samples and is defined as eq. (5):

$$recall = \ TP \ / \ (TP + FN)$$

(5)

The f1 score (or hereafter also described as "accuracy") is defined as the arithmetic average of the precision and the recall,
with its best value at 1 and its worst score at 0, with equal contribution of recall and precision (eq. 6):

$$f1 = 2 \times (precision \times recall/(precision + recall)$$

(6)

Note that in the current case of binary classification, positive and negative labelling can be inverted so that f1 scores can be
calculated both for positive and negative cases. The accuracy is then calculated as the harmonic mean of the positive and
negative f1 scores. The balanced accuracy on the other hand is defined as the arithmetic mean of the positive and negative
recall. We decided to use several metrics for the evaluation of the classification predictions: first, a confusion matrix was
calculated from which the precision, the recall and the f1 scores were derived. The ratios of positive to negative in the dataset
were closely monitored for each of the evaluations, and both the accuracy (the harmonic mean of the f1 scores for positive and
negative) and the balanced accuracy (the arithmetic mean of the recall) were calculated. When the dataset was imbalanced, a
focus was put on the balanced accuracy to closely monitor the performance of the prediction of positive cases.

## 3 Results

### 3.1 Ceilometer backscatter profile analysis: the 6th of January 2019 case study

430                                                  < Figure 5 here >

To illustrate the disparity between the various cloud phase retrievals, observations from the 6th of January 2019 were selected
as an example, as these included both low and higher clouds, with SLW present both within thick clouds or isolated from thick
clouds. In Figure 5, one can see the calibrated attenuated backscatter from the ceilometer (Fig. 5a), the reference cloud phase
mask combining radar and lidar (Fig. 5b), the depolarisation ratio from the RMAN lidar (Fig. 5c), and the cloud phase
attribution from two retrievals following T19 (Fig. 5d), and G22-Davis (Fig. 5e). The G22-Davis model was trained on a
different set of the data, which excluded the 6th of January 2019. During the first part of the day, until around 11:30 UTC, the
radar-lidar cloud phase mask shows little occurrence of SLW or mixed phase, except at the very beginning of the day (the first
10 min) and around 2:00 UTC. In the second part of the day, clear horizontal bands of SLCC can be observed, including times
with clouds or precipitation below the SLCC bands. The ceilometer backscatter (Fig. 5a) showed distinct signal patterns for
the first and second part of the day, but visually it remains difficult to clearly distinguish strong backscatter during the first
part of the day, that could indicate the presence of SLCC.


< Figure 6 here >

Figure 6 shows five selected vertical profiles of attenuated backscatter (Fig. 5a) for that day, chosen to illustrate the diversity
in attenuated backscattering signal profiles. It also shows the theoretical molecular backscatter at the wavelength of the
ceilometer, the identified peaks, as well as their average properties. For the profiles A, D and E, SLW or mixed phase were
identified in the reference radar-lidar mask. These three backscatter profiles presented common characteristics, such as a
narrow peak (low value of the peak width at mid-height), relatively high values of the attenuated backscatter (above $10^{-4}$ $m^{-1}$
$sr^{-1}$), and high prominences (high values of the difference between peak value and the surrounding baseline). Conversely, peaks
B and C present much wider peaks (higher values of peak width at mid-height), and smaller values of the prominence and were
classified as ice.

< Figure 7 here >


In Figure 7, the average peak properties for which peaks had been identified for the 6th of January 2019 are presented as joint
distributions using kernel density estimation plots with peak β value as the y-axis. We chose to use peak value as the y-axis as
this is the most discriminant peak characteristic, and by analysing scatter plots, we can visually observe clustering patterns. In
Figure 7, a single ß profile can generate multiple datapoints if several peaks were observed for that profile. The points A, D
and E were well clustered within the same region in Fig. 7a and Fig. 7b, corresponding to high values of ß at peak, low values
of the peak width and peak width height. Peaks A and E also appeared in the same region in Fig. 7c. Generally, SLW was
observed when a single peak was observed for the ß profile. This was not always the case, as sometimes, SLCC can be observed
when several peaks are present: for instance, on the 6th of January, the isolines showed that several instances of SLCC were
observed when two peaks were present, but this dropped dramatically for three or four peaks. For that specific day, the scatter
plot with peak altitude (Fig. 7d) shows that SLCC is more frequent at higher altitudes with three clusters around 1000 m, 2000
m and 3500 m AGL, corresponding to the spatial organisation of SLCC (Fig. 5b), while non-SLCC peaks are far more frequent
at lower altitudes (cluster located between 0 and 1000 m AGL). These non-SLCC peaks are associated to low-level clouds as
no fog was identified over that three-month period.

The data shown in Figures 5, 6, 7 demonstrate that peak average characteristics exhibit very specific features that could be
used to detect the occurrence of SLCC. The original approach from T19 was already skilful in identifying SLCC. For the

second half of the day on January 6[th], 2019, there was a very good match between the cloud mask based on T19 (Fig. 5c) and the reference cloud mask (Fig. 5b). Conversely, for the first half of the day, the T19 approach identified multiple SLCC regions within the cloud producing a speckle pattern of SLCC. This was not observed by the radar-lidar observations and algorithm

and was thus probably wrongly labelled by the T19 approach. Our new approach based on the use of average peak characteristics and a dedicated trained algorithm (G22-Davis) showed a great improvement in the retrievals: the second half of the day remained like the retrieval of T19, although labelling thicker bands of SLW utilising the peak width property. These thicker horizontal bands of SLCC were more in line with the radar-lidar reference, which showed occurrence of SLW or mixed phase of about the same thickness. For the first half of the day, G22-Davis outperforms the T19 approach, by removing the

spurious speckle patterns while keeping the correct detection of SLW at the very beginning of the day, also observed in the radar-lidar cloud mask (profile A in Fig. 6a). The SLW around 2:00 UTC found in the radar-lidar cloud phase mask was also retained by our new technique. An occurrence of SLCC at around 5:00 UTC was present in our retrieval but not in the reference radar-lidar cloud mask. This may be due to an error in our retrieval, an error in the phase assignment in the lidar product, or different observations made by the lidar, radar and ceilometer at that timestep. Nonetheless, G22-Davis performed very well:

computed accuracies for that day (6[th] of January 2019) using the radar-lidar mask as the reference were equals to 0.84 for our new data-driven retrieval, compared to an accuracy score of 0.65 with the T19 approach. In the next section, we evaluate the performances of T19, a data-driven threshold approach and G22-Davis for our full dataset covering almost three months of data during the Southern Hemisphere summer.

**3.2 Evaluation of retrievals for the PLATO period**

In Figure 8, similarly to what was presented in Figure 7, the average peak properties for which peaks had been identified are presented as joint distribution plots with peak $\beta$ value as the y-axis, but this time for the full dataset. The two-dimensional distributions are like that of Figure 7, showing that the results of the case of January 6[th] can be extrapolated to the full dataset. The value of $\beta$ at peak versus peak prominence is not presented as these are directly correlated. The value of $\beta$ at peak is

correlated to the peak width height, with some differences. The value of $\beta$ at peak is clearly the best feature to separate the SLCC-labelled datapoints (True) to the non-SLCC datapoints (False). The other features, e.g., peak width, number of peaks and peak altitude are weaker discriminants, but they reveal that non-SLCC datapoints (False) are much more widespread than SLCC datapoints (True), showing that extreme values of the features are usually associated with the non-occurrence of SLCC. Typically, many peaks (> 2 bins or 100 m), or very wide peak widths (> 3 bins or 150 m) for each profile are associated to

non-SLCC. The presence of a dense concentration of non-SLCC datapoints at lower altitudes (< 1000 m) shows that SLCC is usually not observed at these low altitudes above ground level.

< Figure 8 here >

As discussed previously, various sub-sampling strategies were considered to compare the ceilometer mask and the reference radar-lidar cloud mask. The comparison between the T19 algorithm and the reference dataset using the various subsampling strategies showed minimal variability with accuracies varying between 0.84 and 0.85 for threshold consecutive values of 3, 4 and 5 and instantaneous comparisons. For the full dataset, the accuracy score was 0.84 (or 0.85 depending on the subsampling

strategy used for the comparison), while for the dataset with peaks only, the accuracy was 0.72.

Based on the clustering as observed in Figure 8 for the full dataset, we decided to implement a second classification using empirical thresholds for each of the peak features. This has the advantage of not having to train and test a model and

circumnavigates the need for a high-resolution reference dataset (although the arbitrary thresholds are based here on the cluster plots where the labelling has been done using the reference dataset). The arbitrary thresholds that we used to label a timestep as "SLW" were for peak features such as: value of $\beta$ at peak $> 5$ x $10^{-5}$ $m^{-1}$ $sr^{-1}$, peak width $< 4$, peak width height $> 40$ x $10^{-6}$ $m^{-1}$ $sr^{-1}$, peak prominence value $> 60$ x $10^{-6}$ $m^{-1}$ $sr^{-1}$ and the total number of peaks $< 3$. We evaluated the arbitrary threshold approach on both the full dataset and the dataset with only identified peaks. For the full dataset, the accuracy score was 0.89,

while for the dataset including peaks only, the accuracy was 0.76.

Our G22-Davis model was trained and tested using a 3-fold stratified cross validation approach as described previously. The model was trained and tested for both the full dataset, and the dataset containing only timesteps for which peaks had been identified. For the full dataset, the best testing accuracy score was 0.91 (with learning rate = 0.005, max depth = 12, child

weight = 8) for an average training accuracy score of 0.95. This is an improvement of 0.07 as compared to the accuracy of T19, which was equal to 0.84. For the dataset for which peaks were identified, the total dataset was made of 11,327 datapoints. The best testing score was of 0.81 (with learning rate = 0.01, max depth = 12, child weight = 8), for training accuracy scores (or f1) of 0.94. This is a substantial improvement of almost 0.1 as compared to the accuracy of T19, which was equal to 0.72.

**3.3 Evaluating the importance of predictors for G22-Davis**

While our designed, trained and tested extreme gradient boosting model performs remarkably well, we want to understand which of the average profile characteristics are the most important for skilful model predictions. Lundberg et al. (2020) recently proposed an explanation method for tree-based models, building on work based on classic game theoretic Shapley values (Lundberg and Lee, 2017). We implemented this explanation method, also known as "TreeExplainer" (Lundberg et al., 2020)

in its original python version. With TreeExplainer, we were able to provide local explanations for each prediction by calculating their Shapley (SHAP) values. The input features were the same as described for the implementation of G22-Davis,

e.g., the average adjusted peak properties. Figure 9 shows the distribution of SHAP values for these 8 input features, together with their normalised value represented as a colour bar.


< Figure 9 here >

As expected, the value of ß at the peak location was the most important feature to produce accurate predictions: high values of
ß were important to detect the presence of SLCC, while low values of ß were skilful in predicting the absence of SLCC. The
total peak number was the second feature of importance, with low values of the total peak number contributing to better
predictions of the presence of SLCC. In fact, very well-defined horizontal bands of SLCC showed these typical characteristics
with a single narrow peak of high values. Peak prominence had a similar SHAP values distribution pattern as peak value, with
less defined clusters. This result is consistent with the joint distribution plot of peak value versus peak prominence (not shown).
Low values of peak width were important in the prediction of the presence of SLCC, showing the importance of the peak shape
(narrow) in indicating the presence of SLCC. Low values of the peak width height tend to also help predict the presence of
SLCC. Conversely, peak altitude and peak temperature are associated with average SHAP values close to 0, indicating that
these two features were not important in the production of accurate predictions. Given the low importance of air temperature
for the accurate prediction of SLCC, we could consider not using that feature as an input to G22-Davis and removing our
dependency on ERA5 or other NWP inputs. However, at Davis, we might likely be in the specific case where air temperatures
are often too negative to produce liquid water droplets (other than supercooled) as seen in the T19 cloud phase classification.
For other climates with higher air temperatures, the air temperature feature might be more relevant. While we saw a tendency
of SLCC to be located within a preferred range of altitudes (between approximately 1000 and 4000 m ASL), this range was
too wide to make altitude of the peak a useful criterion to help identify SLCC. Low values of peak altitude however seemed
to help identify conditions with the absence of SLCC (as indicated by the negative and blue SHAP values for the peak altitude).

## 3.4 Climatology of SLCC and other clouds over one year

< Figure 10 here >

Here, we apply the T19 and G22-Davis techniques to characterize the frequency of occurrence of SLCC using a full year of
ceilometer data (November 2018 to Octover 2019). This full year of SLCC retrieval (T19 and G22-Davis) provides a unique
opportunity to document and analyse seasonal variations in the occurrence of SLCC and clouds without SLCC, as well as the
presence of fog. In Figure 10, SLCC retrieved by T19 had larger frequencies of occurrence for most of the year except for the

months of September and October 2019, whereas SLCC retrieved from G22-Davis were slightly more frequent. As discussed previously, the speckle patterns seen in T19 and absent in G22-Davis is mostly responsible for that difference between the two retrievals. Clouds other than SLCC have very similar frequencies of occurrence for T19 and G22-Davis since the detection technique for these other clouds is the same in both methods, and the differences are only due to the few datapoints that were

classified as SLCC in one method and not in the other. Overall, the frequency of SLCC over the full dataset is of $0.29 \pm 0.06$ for T19 and of $0.24 \pm 0.05$ for G22-Davis, while the frequency of clouds other than SLCC is of $0.29 \pm 0.10$ for T19 and of $0.27 \pm 0.10$ for G22-Davis. The frequency of fog is of 0.006 for the full dataset. Hamalainen et al. (2020) reported frequencies of occurrence of SLCC between 12% and 30% depending on the location of their observations across Finland (covering the furthermost southern and northern parts of the country) and height above ground, during the Northern Hemisphere winter

period. Ricaud et al. (2020) measured frequencies of occurrence of SLW of up to 50% in December and January at Dome C, Antarctica but their criteria for defining occurrence of SLW during a day was if the SLW clouds were present at least 1h during the day. They also reported much lower frequencies of occurrence during March and April down to 10% or less. Cossich et al. (2021) showed from their observations that mixed phase clouds were rare or inexistent in Austral winter on the Antarctic plateau at Concordia, and up to 11% during the Austral summer months.


**4 Discussion and conclusions**

In this study, we have demonstrated that the presence of supercooled liquid water can be detected using the measured attenuated backscatter signal from a ceilometer, even in the absence of information on the signal depolarisation. We utilised coincident

observations from a W-Band radar and a depolarisation lidar to build a reference cloud phase mask that was used as a benchmark to compare to our ceilometer-retrieved supercooled liquid water detections. Utilising the ceilometer data and an existing approach proposed by T19, we obtained an overall accuracy of 0.84 and an accuracy for days with detection of strong backscatter signals of 0.72. We then developed an enhanced method, e.g., G22-Davis, utilising the benchmark dataset of cloud phase observations to develop, train and test an extreme gradient boosting model.


Utilising G22-Davis, we increased the overall accuracy of correctly identifying SLCC layers to 0.91 and the accuracy for days with a detection of strong backscatter signals to 0.81. G22-Davis greatly improved detection accuracy, for the cases where multiple peaks in the backscatter were observed and were erroneously classified as SLCC by the method from T19. The most important input features were the value of the backscattered signal at the peak, followed by the total number of peaks within

that profile. In the current approach, we considered each profile (or timestep) independently from one another (although we did consider all points together in the statistical analysis and in the model development). Alternative or complementary to G22-Davis, data-driven approaches such as Recurrent Neural Networks (Long Short-Term Memory or Gated Recurrent Unit) could

consider the spatio-temporal patterns of peak properties to predict the occurrence and location of SLW at the following timestep. Finally, in the absence of radar-lidar mask to train and test a model like in G22-Davis, there remains the possibility of using the classification approach based on thresholds derived from peak characteristics joint distributions. This method should be assessed for different periods at Davis and for other locations than Davis, to test of these threshold values are widely applicable. The frequencies of occurrence of SLCC reported here of $0.24 \pm 0.05$ is within the range of observations collected in Finland and at Dome C, although the definitions used to report frequencies differ depending on the authors. We showed that at least 15% of SLCC were observed in the Austral winter, while Cossich et al. (2021) reported rare or inexistent mixed phase clouds on the Antarctic plateau in winter. This is probably due to moister and relatively warmer environments on the coastal fringes of the Antarctica continent compared to the environment of the drier and colder plateau.

Ceilometers are relatively low-cost ground-based active atmospheric remote sensing tools as compared to weather radars or depolarization lidars. They are commonly deployed at aerodromes but also at other operational or research atmospheric monitoring facilities. Here, we showed that the attenuated backscatter signal can be utilised to detect supercooled liquid water, thus broadening the observational capabilities of such instruments, for regions where observations are scarce, like Antarctica. The present work is the first of its kind utilising a benchmark radar-lidar cloud phase mask to train a dedicated model to detect supercooled liquid water from the ceilometer backscatter only. It will be important to test the application of the same approach elsewhere, especially for current monitoring sites or for historical data including the same set of instruments presented here, that is weather radar, depolarisation lidar and ceilometers. Our approach was developed for a polar environment, and it would be important to see how the developed technique transfers to regions at mid or low latitudes. One important aspect of our approach is that each locally trained model will provide a given cloud phase retrieval, and various training sets will give various cloud phase retrievals. We labelled our model G22-Davis and following that logic, we can imagine for example, G22-Casey as another model trained on data collected at Casey station. It will be important in future work to evaluate the difference between model retrievals based on various training sets for the same applied dataset. Given these constraints, our other approach proposed in this study, using empirically defined thresholds on peak characteristics could provide a benchmark cloud phase model to refer to, to evaluate each of the G22 trained models.

Ground-based observations of supercooled liquid water are complementary to spaceborne observations. Satellite detection of supercooled liquid water suffers from the attenuation of the signal in the lower layers, and from a lower spatial and temporal resolution. The combination of satellite and ground observations has the potential to improve cloud phase products. Knowledge of the cloud phase including supercooled liquid water at high-resolution can help develop and validate icing algorithms, with the objective of predicting aircraft airframe icing potential (Morcrette et al., 2019) or predicting the potential icing of wind turbines for wind production (Hamalainen et al., 2020). Detection capabilities developed in this paper will enable

important studies to examine the seasonal variability of the occurrence of SLCC and to develop aircraft icing potential nowcasting capabilities.

**Code and data availability**

The ALCF is open-source and available at  https://alcf-lidar.github.io (last access: 2 May 2022) as well as permanent archive of code and technical documentation on Zenodo at https://doi.org/10.5281/zenodo.4411633 (Kuma et al., 2021). A tool for
converting Vaisala CL31 and CL51 data files to netCDF cl2nc is open-source and available at https://doi.org/10.5281/zenodo.4409716 (Kuma, 2020a). The observational data (ALCF-processed netCDF ceilometer files and the radar-lidar mask netCDF files) are available on Zenodo at https://doi.org/10.5281/zenodo.5832199 and will also be available on the Australian Antarctic Division datacentre. The ERA5 data are available through the Copernicus data portal at https://cds.climate.copernicus.eu (last access: 2 May 2022). The new G22-Davis ceilometer algorithm described herein as well
as the original T1 algorithms are in the process of being included in ALCF and will therefore be open-source and publicly available.

**Author contribution**

AG performed the ceilometer code development and overall data analysis. AP and SA analysed the radar and depolarisation lidar data and produced the radar-lidar cloud mask. AP, SA, and AK provided regular scientific inputs on the analysis. AP,
SA, AK, and AMD designed the experimental setup at Davis to gather ground-based observations. PK developed the code to pre-process the ceilometer data and provided scientific inputs on the ceilometer data analysis. AG prepared the manuscript with contributions from all co-authors.

**Competing interests**


The authors declare that they have no conflict of interest.

**Acknowledgments**


Deployment of the instrumentation to Davis, and the contribution of S.P. Alexander to this project, was supported by the Australian Antarctic Division through Australian Antarctic Science Projects 4292 and 4387. We thank Ken Barrett, Chris Young, Derryn Harvie, Mike Hyde and Angus Davis for their assistance in preparation, installation, and commissioning of the

instruments at Davis. Thanks to Dr Hanh Nguyen from the Australian Bureau of Meteorology for her contribution in merging

RMAN lidar and W-Band radar products. The open-source libraries Pandas (The pandas development team, 2010), Numpy (Van der Walt et al., 2011), Scipy (Virtanen et al., 2019), matplotlib (Hunter, 2007), netCDF4 (Rew and Davis, 1990), scikit-learn (Pedregosa et al., 2011), XGBoost (Chen et al., 2015) in the python programming language (Rossum, 1995) were used to develop and implement the code to process the data. The python implementation of TreeExplainer is available at https://github.com/slundberg/shap (last accessed December 10th, 2021). Finally, thanks to the two anonymous reviewers

and the associate editor for their comments that helped significantly improve the quality of the original manuscript.

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

**Figures**

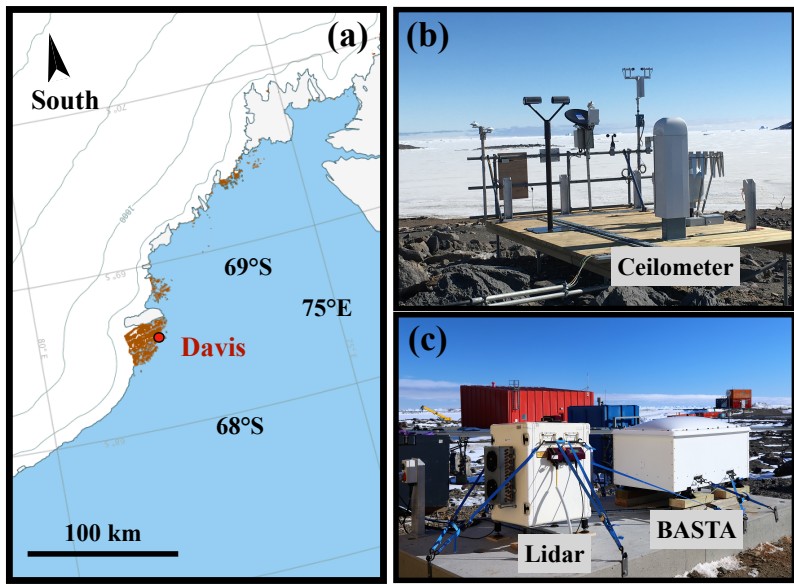

Figure 1: (a) Location of Davis in East Antarctica; the basemap was extracted from the Antarctic Digital Database from the Scientific Committee on Antarctic Research (SCAR) from the British Antarctic Survey; rocky outcrops are shown in brown and elevation as contour lines every 500 m; (b) Photograph of the Vaisala CL51 ceilometer (foreground) installed on the meteorological platform together with other instruments not used in this study; credit: Andrew Klekociuk, Australian Antarctic Division; (c) Photograph of the W-Band radar (BASTA) and the Raman depolarisation lidar mounted on a dedicated concrete slab; credit: Simon Alexander, Australian Antarctic

Division.

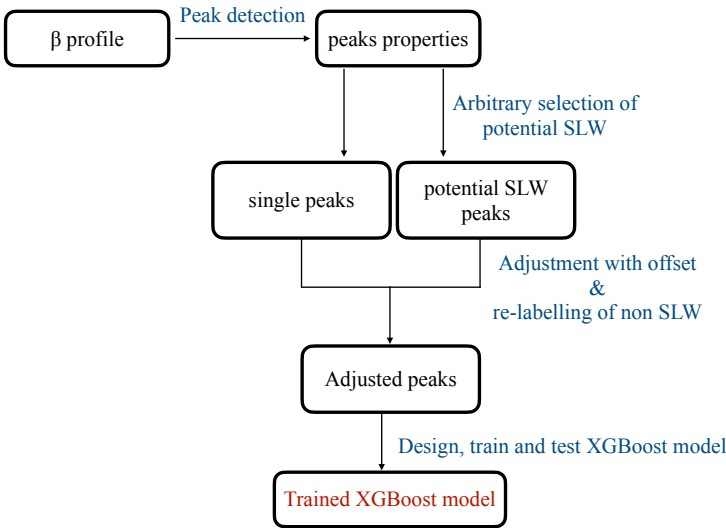

Figure 2: Flowchart describing the pre-processing stage in our new algorithm, e.g., G22_Davis: data preparation, model development, training, and testing.


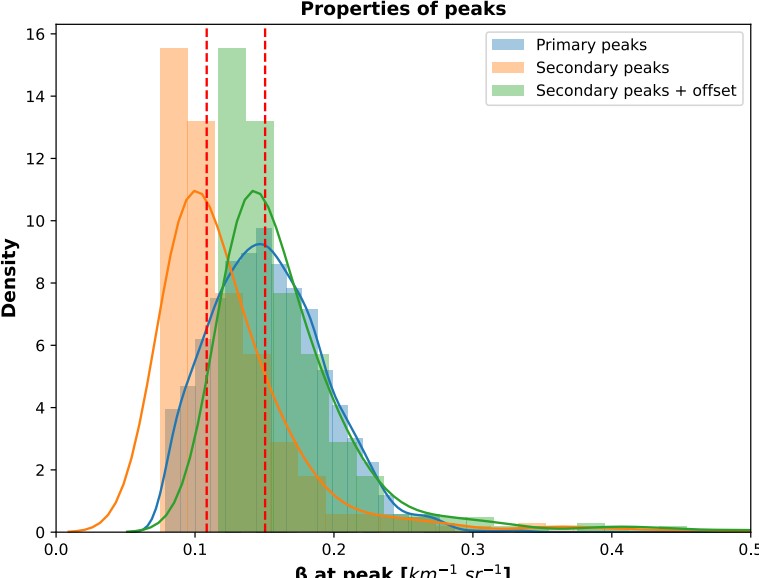

Figure 3: Distributions and Kernel Density Estimates of values of attenuated backscatter for identified peaks. Primary peaks are labelled in blue (3,727 datapoints), while profiles including secondary peaks (peak numbers equal to 2, 3 or 4) are shown in orange (570 datapoints, including 539 datapoints with a peak number = 2, 26 datapoints with a peak number = 3 and 5 datapoints with a peak number = 4). Vertical dashed red lines indicate the median values of primary and secondary peak distributions. Adjusted secondary peaks (secondary peak attenuated backscatter values + offset) are shown in green.



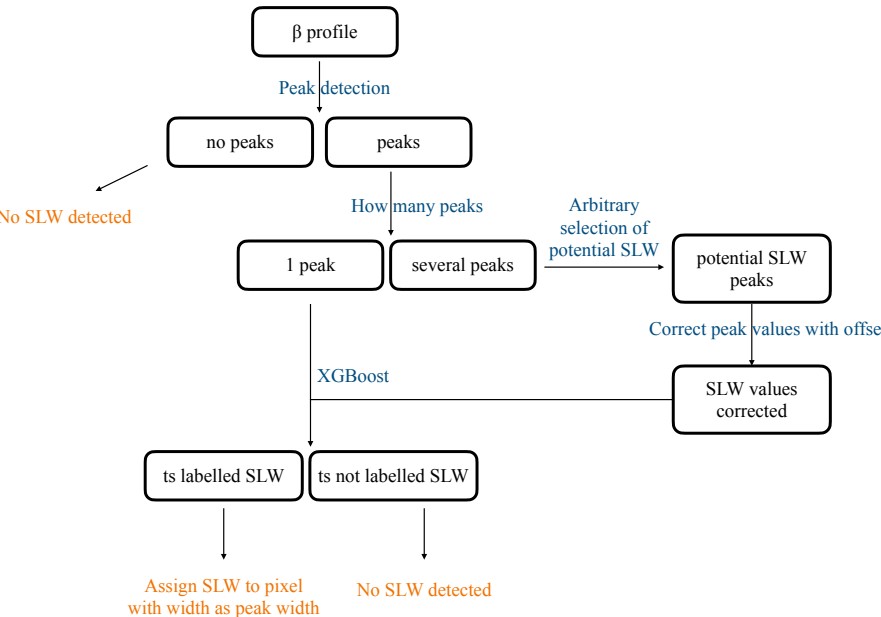

Figure 4: Flowchart describing the second phase of G22-Davis: SLW detection as part of the cloud phase mask algorithm ("ts" is for time-step).


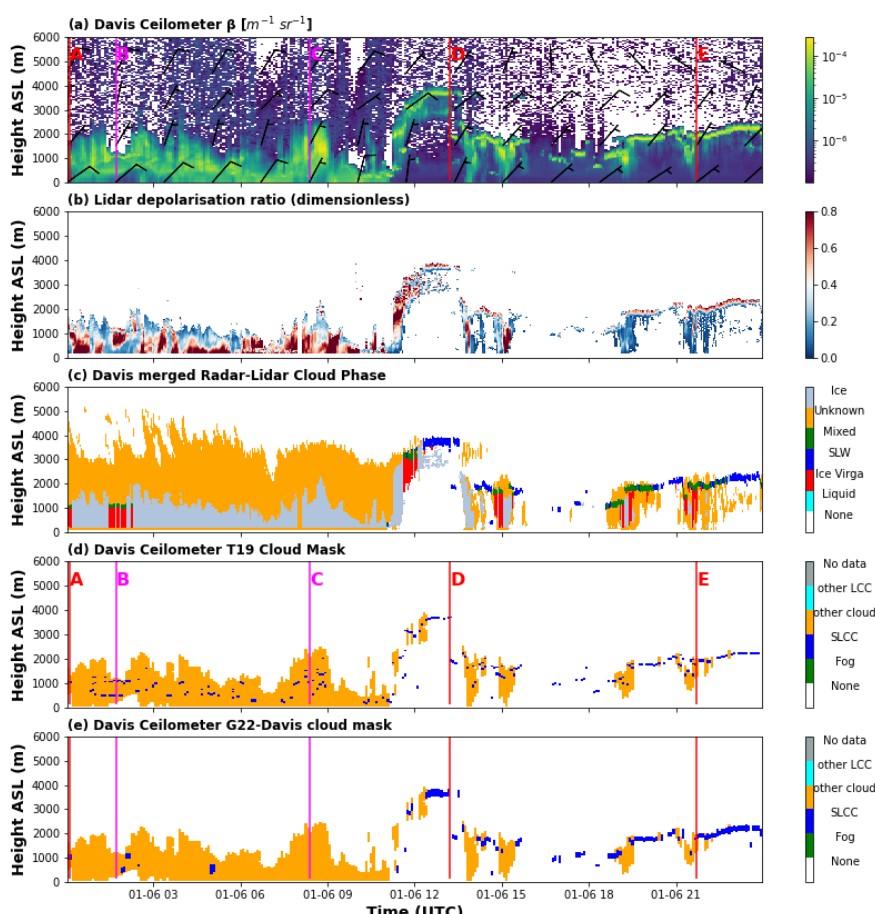

Figure 5. For the selected day of the 6[th] of January 2019: (a) Calibrated attenuated total volume backscatter and wind barbs (wind data extracted from ERA5); (b) Depolarisation ratio; (c) Cloud phase mask following the algorithm adapted from Alexander and Protat (2018); (d) Cloud mask based on T19; (e) Our new cloud mask based on G22-Davis. Selected vertical profiles have been selected (A, D, and E in red correspond to identified SLCC, while B and C in fuchsia correspond to non-SLCC occurrences).

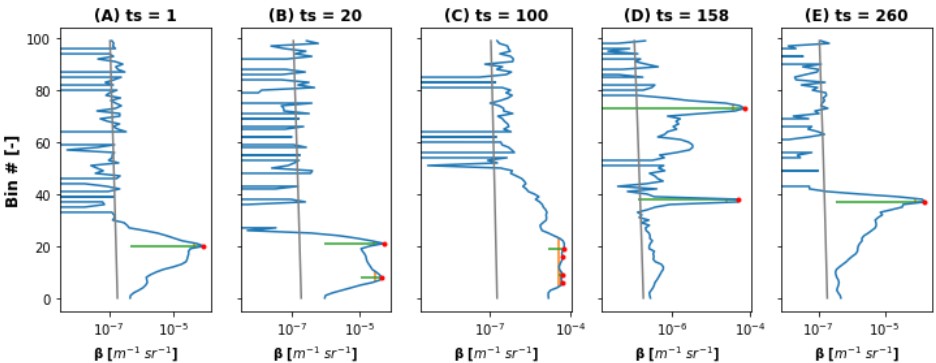

Figure 6. For the 6th of January 2019: selected vertical profiles of backscatter (A, B, C, D, E). Timesteps are indicated as "ts" and can be converted to time by multiplying by 5 min. Peak are identified with a red dot, peak widths with a vertical orange line and peak prominences with a horizontal green line. The baselines (lowest points on the green lines) were calculated as the lowest contour lines around the peak. To identify the peak characteristics, we used the signal processing tools of the python library SciPy. The theoretical molecular backscattering was computed following the equation as found in Kuma et al. (2021) and is shown on each subpanel as a solid grey line.


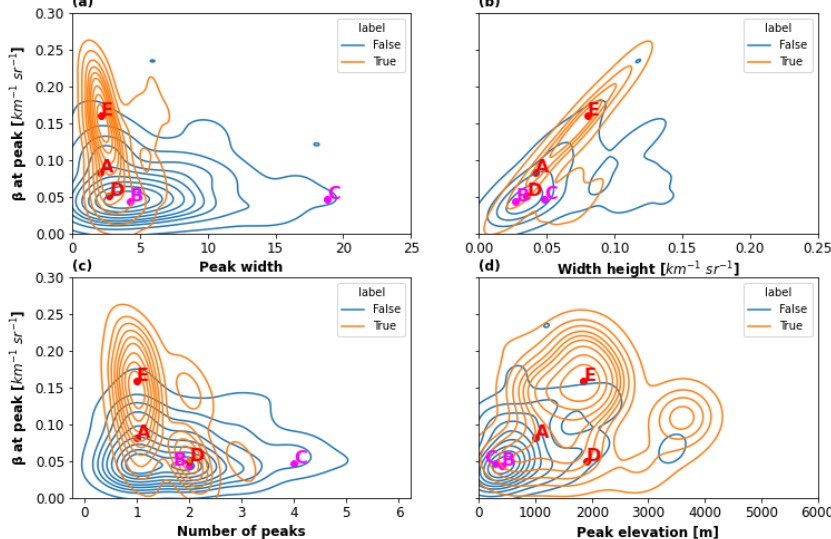

Figure 7. Joint distribution using kernel density estimation plots showing the distribution of peak average properties for the 6th of January
2019: (a) attenuated backscatter versus peak width; (b) attenuated backscatter versus peak width height; (c) attenuated backscatter versus the number of peaks within the profile; (d) attenuated backscatter versus peak altitude above ground level. Isolines are shown with a spacing of 0.1 and the label "True" corresponds to SLW or mixed phase observed by the reference radar-lidar mask, while "False" indicate

no detection by the reference mask. Selected vertical profiles from Figures 5 and 6 are also shown with red dots (A, D and E: SLW occurrences, e.g., "True") and fuchsia dots (B and C: non-SLW occurrences, e.g., "False").


000    Figure 8. Joint distribution using kernel density estimation plots showing the distribution of peak average properties for full period of the PLATO observations (November 2018 to February 2019: equivalent to a total of 11,327 datapoints): (a) attenuated backscatter versus peak width; (b) attenuated backscatter versus peak width height; (c) attenuated backscatter versus the number of peaks within the profile; (d) attenuated backscatter versus peak altitude above ground level. Isolines are shown with a spacing of 0.1 and the label "True" corresponds to SLW or mixed phase observed by the reference radar-lidar mask, while "False" indicate no detection by the reference mask.

005

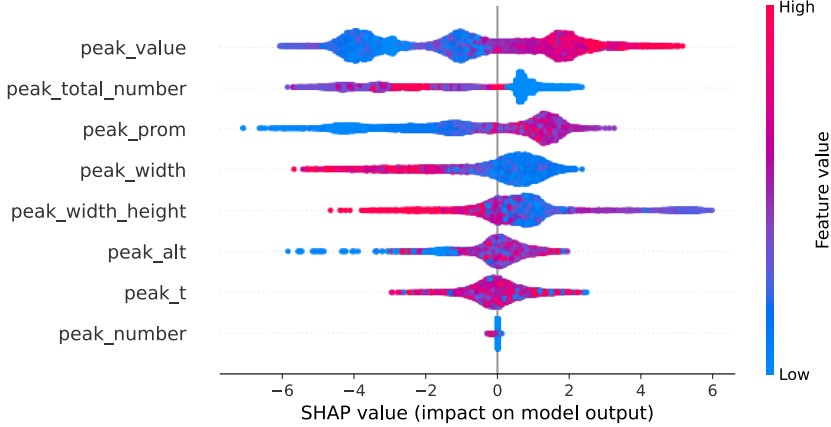

Figure 9: Distribution of Shapley values (SHAP values) calculated by TreeExplainer applied to G22-Davis. Features (for meaning of abbreviations refer to Table 1) are ranked from the most important (top of the list, e.g., peak_value) to the least important (bottom of the list, e.g., peak_number). The normalised feature values are shown with a blue to pink gradient as indicated by the right-hand side colour bar.

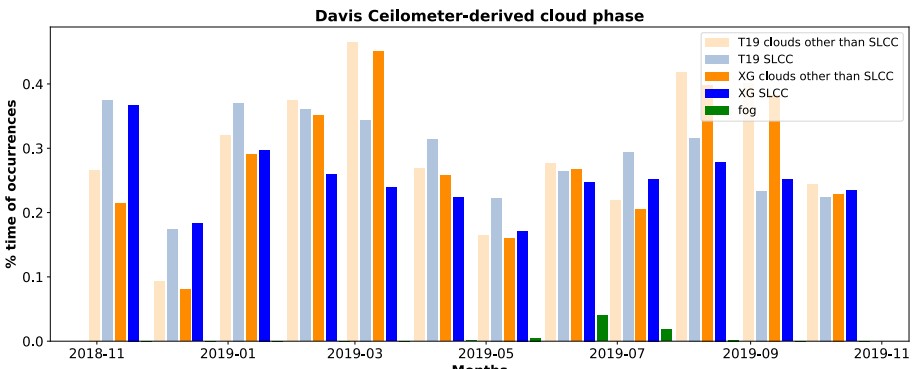

Figure 10: Cloud phase occurrences (SLCC or clouds other than SLCC) as retrieved using T19 or G22-Davis. Fog occurrences are also shown.