# Peer review of "Detection of supercooled liquid water containing clouds with ceilometers: Development and evaluation of deterministic and datadriven retrievals"

_Atmospheric Measurement Techniques, 2022_

## Referee Comment (RC1)

Review of amt-2022-10
"Detection of supercooled liquid water clouds
with ceilometers: Development and evaluation of
deterministic and data-driven retrievals"

March 2022

**Overall comment**

New algorithm has been developed to detect supercooled liquid water (SLW) by just using ceilometers. New method is compared with the existing method to observe SLW with more extensive observational data set and existing algorithms to identify SLW with only ceilometer. Authors have developed new method, utilising machine learning, that perform better compared to the existing algorithm for simple instrumentation. Manuscript is well written and mainly clear - some specific clarifications are requested in the followed section in detailed comments. I am suggesting this manuscript will be accepted after minor revision (mainly clarifications).

**Detailed commets**

How does authors see the potential of using this method elsewhere i.e would it require location-specific training set to detect SLW? Would be interesting to see test results for other location with / without specific training set. In addition, two models - trained with location specific data - could give different results on the same attenuated backscatter profile, right? Could authors elaborate on these aspect a bit in the manuscript. Would this cause some problem in some applications? Is this something to be accounted for?

line 49: Please add reference for: "Typically, a depolarization ratio below 10% is characteristic of SLW clouds, while higher values are produced by ice particles"

line 147: it is stated: "(2) remove noise by applying a noise removal algorithm and subsampling the data to 5 min, 50 bins;"
Could you please describe what kind of noise removal is applied – how it is done

– and same for bin sampling. More information is needed.

line 148: "(3) calibrate the attenuated backscatter using the approach of Hopkin et al. (2019)"
Please clarify if this is done after subsampling and if subsampling is generating some effect to the method?

line185: Please clarify how ice virga is defined?

line 257: It is stated that "the width of the peak must be $< 4$,"
It remains unclear, what units are in case of 4? Range gates (4*50m)? Please clarify, for example stating ", corresponding to X meters".

line 249: "extinction other than molecular in the lower levels, cannot be directly compared in terms of backscatter values to peak"
Can you please clarify the terminology usage, why "extinction" and not "attenuation"? It is also stated that there is no other extinction than molecular – how about effect of aerosols?

Line 258: Does this "multiple peaks" -group consist all cases where number of peaks $> 1$? Did you check if there is any difference between beta value in cases of 2nd peak and 3rd peak? Is "multiple peaks" usually only 2 peaks. Will there rise some implications in case of 3 peaks?

Lines 265-270 "The difference between the median value of the single peak distribution and the multiple peak distribution can be calculated and is equal to 4.20 x 10-5 m-1 sr-1 ."
It remains unclear how this was calculated, please clarify. Is there difference between 2nd and 3rd peaks (see previous comment)?

Section 2.3 Enhanced data-driven ceilometer cloud phase mask
More information is needed in this section. Please clarify and describe:
- line 282: what are these "learners"? Please clarify. - How are the training and validation data sets selected? How long training set? How long validation data?
- Can you please tie these ML world terms into observations / data used. Many aspects remains unclear for the people unfamiliar with machine learning and it would be impossible for people to reproduce this algorithm. Any comment on this aspect?

figure 5: would it be possible to show the depolarization values in this plot? Would be interesting to see the data especially in the first half of the data period when T19 method gives lots of SLW detection and reference not.

line 435: How good is the reference in the first place? This reference method could be described in more detail as it remain unclear how the reference combination algorithm is working. Is this method seen as reliable reference? Please offer some references.

line 538: "raw" means different things for different people. I suggest to clarify, do you mean the data in output files or "attenuated backscatter" - meaning the data after calibration. Is it necessary to calibrate the signal before use?

line 539: Just out of curiosity: how many ceilometers there are in Antartica where no other potential devices are installed?

line 561: "The new ceilometer algorithm described herein has been developed at the Bureau of Meteorology and is not publicly available."
For what use this algorithm is developed if it is not publicly available and cannot be reproduced based on this manuscript?

---

## Author Comment (AC1)

**amt-2022-10**
**" Detection of supercooled liquid water clouds with ceilometers: Development and evaluation of deterministic and data-driven retrievals"**

**Authors' response to comments from reviewer #1**

**2 May 2022**

We would like to thank the editor and the two reviewers for their very constructive comments on our manuscript. We received genuine insights, which have significantly contributed to increasing the manuscript quality and potential impact. To improve the clarity in our responses we have numbered the reviewers' comments: for example, the comment 1 from reviewer 1 is listed as R1C1 and will refer to these comments as such in the following.

Based on some of the comments from the reviewers and late feedback from our co-authors, we also made small additional changes to the original manuscript:

(1) Following internal discussion between the co-authors, we realised that our initial definition of Supercooled liquid clouds (SLW) was misleading as it included both mixed-phase and SLW clouds. With our technique, and the technique from T19, we can detect clouds containing SLW, e.g., both SLW clouds and mixed phase clouds, as we cannot distinguish between the two. We therefore decided to replace instances where we referred to both SLW and mixed phase as "Supercooled Liquid water Containing Clouds" or SLCC, including in the title.

(2) We have incorporated late feedback from one of our co-authors; these were mostly typo and added precisions in the manuscript and figures.

(3) We have applied our two retrievals to additional data from the same instrument (the ceilometer) that covers a full annual cycle including austral winter months. We have included a new section 3.4 in the manuscript and a new Figure 10. We think that this will add value to the paper, showing how this new retrieval can produce climatology of supercooled liquid water containing clouds in regions such as Antarctica where observations are scarce. We also added a few sentences putting these results into perspective and in the light of other observations in Antarctica and elsewhere.

(4) We made some cosmetic modifications in the Figures: In Figure 1a, the arrow was wrongly labelled "North", we removed the label to Progress 3 and removed the bathymetry. In Figure 5, we changed the x-axis units. Figures 7 and 8 saw the unit of the axis changes to km instead of m to simplify the axis unit labels.

Reviewer #1

Overall comment

New algorithm has been developed to detect supercooled liquid water (SLW) by just using ceilometers. New method is compared with the existing method to observe SLW with more extensive observational data set and existing algorithms to identify SLW with only ceilometer. Authors have developed new method, utilising machine learning, that perform better compared to the existing algorithm for simple instrumentation. Manuscript is well written and mainly clear - some specific clarifications are requested in the followed section in detailed comments. I am suggesting this manuscript will be accepted after minor revision (mainly clarifications).

Detailed comments

How does authors see the potential of using this method elsewhere i.e would it require location-specific training set to detect SLW? Would be interesting to see test results for other location with / without specific training set. In addition, two models - trained with location specific data - could give different results on the same attenuated backscatter profile, right? Could authors elaborate on these aspect a bit in the manuscript. Would this cause some problem in some applications? Is this something to be accounted for?

R1C1: Thanks, this is indeed a very important aspect, and we may have overlooked this in our discussion. We have now added this paragraph to the discussion section (lines 973-980):

"One important aspect of our approach is that each locally trained model will provide a given cloud phase retrieval, and various training sets will give various cloud phase retrievals. We labelled our model G22-Davis and following that logic, we can imagine for example, G22-Casey as another model trained on data collected at Casey station. It will be important in future work to evaluate the difference between model retrievals based on various training sets for the same applied dataset. Given these constraints, our other approach proposed in this study, using empirically defined thresholds on peak characteristics could provide a benchmark cloud phase model to refer to, to evaluate each of the G22 trained models. "

Work is currently underway utilising observations in NZ to evaluate how the current approach and model could transfer to these datasets.

line 49: Please add reference for: "Typically, a depolarization ratio below 10% is characteristic of SLW clouds, while higher values are produced by ice particles"

R1C2: The reference Ricaud et al. (2020) (already listed in the references) has been added to that sentence.

line 147: it is stated: "(2) remove noise by applying a noise removal algorithm and subsampling the data to 5 min, 50 bins;"
Could you please describe what kind of noise removal is applied – how it is done and same for bin sampling. More information is needed.

R1C3: Thanks, we added more details as follows (lines 159 to 153):

" "(2) remove noise by applying a noise removal algorithm and subsampling the data to 5 min, 50 bins; The noise removal is done by estimating the distribution of noise at the highest available range and subtracting the mean of the distribution (scaled by the square of range) from all bins in the column. In the cloud masking, the standard deviation of noise is considered when determining if a bin is cloudy. By default, five standard deviations are subtracted from the value before the cloud mask threshold is applied. This is done to prevent false positives with sufficient probability; Subsampling is mostly done to improve signal-to-noise ratio. The cloud masking usually benefits from subsampling to 5 min intervals and 50 m vertical resolution, because it decreases the number of misclassified bins;"

line 148: "(3) calibrate the attenuated backscatter using the approach of Hopkin et al. (2019)"
Please clarify if this is done after subsampling and if subsampling is generating some effect to the method?

R1C4: This is done after subsampling and we have clarified this in the text, it now reads as:

"(3) After noise removal, then subsampling, ALCF performs a calibration of the attenuated backscatter using the approach of Hopkin et al. (2019). In addition to the absolute calibration, the instrument built-in software applies overlap calibration internally. The final pre-processed products were daily netCDF files including the total attenuated volume backscattering coefficient (ß, $m^{-1}$ $sr^{-1}$) at a resolution of 5 min and bin vertical resolution of 50 m."

line185: Please clarify how ice virga is defined?

R1C5: We presume you refer to line 135. We added this sentence:

"… perpendicular channel. Ice is defined to be virga rather than cloud when it exists beneath a SLW layer. It also…"

line 257: It is stated that "the width of the peak must be < 4,"
It remains unclear, what units are in case of 4? Range gates (4*50m)? Please clarify, for example stating ", corresponding to X meters".

R1C6: The sentence now reads: "the width of the peak must be < 4 bins (corresponding to 200 m)". See also same comment from R#2 at R2C11.

line 249: "extinction other than molecular in the lower levels, cannot be directly compared in terms of backscatter values to peak"
Can you please clarify the terminology usage, why "extinction" and not "attenuation"? It is also stated that there is no other extinction than molecular – how about effect of aerosols?

R1C7: Thanks, we indeed made a mistake here in using extinction where it should have been attenuation. 'Attenuation' refers to the loss of (lidar) signal as the lidar signal propagates up through the atmosphere, for example: 'a SLW cloud layer usually fully attenuates a lidar beam, resulting in no signal return'.

In contrast, the extinction is, more formally, the 'particulate extinction co-efficient' which we calculate from solving the lidar equation. Extinction (along with backscatter and lidar ratio) are the properties of the aerosols / clouds themselves.

The presence of aerosols along coastal Antarctica has been demonstrated (as measured by ship campaigns (Humphries et al. 2021, https://doi.org/10.5194/acp-21-12757-2021). Near the coast, there is much less sea salt aerosol than over the warm Southern Ocean, instead aerosols are mainly biological in origin.

However, the extinction by clouds far exceeds extinction from aerosols. For the whole PLATO campaign (3 months of Summer), we only recorded one nice clear aerosol example (as per the below figure that shows depolarisation ratio). The aerosols can be seen at 500-1000m from about 1500 UTC onward. The detection of aerosols during PLATO is limited by the continuous daylight (local midnight is at about 1930UTC), and thus high SNR for most of the summer.

[Figure]

Line 258: Does this "multiple peaks" -group consist all cases where number of peaks > 1? Did you check if there is any difference between beta value in cases of 2nd peak and 3rd peak? Is "multiple peaks" usually only 2 peaks. Will there rise some implications in case of 3 peaks?

R1C8: Thanks, this is a very relevant question, and we indeed failed to address this in the manuscript. This was also raised by Reviewer #2 in R2C12 and R2C13. In the paper, the "multiple peaks" group include peaks within a profile that are either second (in terms of altitude, so after a first peak has been identified), or third or fourth. While the "single peaks" group include all first peaks identified, whether there are multiple peaks in the profile. We now realise this could be confusing, so we propose to change the terminology to:

"Multiple peaks" → changed to → "Secondary peaks"

"Single peaks" → changed to "Primary peaks"

This is changed in the text and figures.

In addition, we also add information about the cases where peak numbers equal to 3 and 4 have been found. The caption of Figure 3 now reads:

"Figure 3: Distributions and Kernel Density Estimates of values of attenuated backscatter for identified peaks. Primary peaks are labelled in blue (3,727 datapoints), while profiles including secondary peaks (peak numbers equal to 2, 3 or 4) are shown in orange (570 datapoints, including 539 datapoints with a peak number = 2, 26 datapoints with a peak number = 3 and 5 datapoints with a peak number = 4). Vertical dashed red lines indicate the median values of primary and secondary peak distributions. Adjusted secondary peaks (secondary peak attenuated backscatter values + offset) are shown in green."

The occurrence of cases with a third peak (26 cases) and a fourth peak (5 cases) are very rare, i.e., 0.6% and 0.1% of the cases, and therefore the number of datapoints is not sufficient to provide a

statistically meaningful distribution to rely on to use the same approach as for the cases with second peaks. We have therefore decided to group all 2nd, 3rd and 4th peak cases into the same group labelled "secondary peaks" and use the properties of that group to adjust the peak values for 2d, 3rd and 4th peaks. This is imperfect, but the only solution we found acceptable given the data constraints.

The paragraph covering this has been modified as follows:

"We call the first group of peaks that don't see lower-level attenuation "primary peaks" and the group of peaks that are higher in altitude above primary peaks "secondary peaks". For primary peaks, data for which SLCC were identified were selected based on the Boolean condition defined using the radar-lidar cloud mask. For secondary peaks, an empirically based set of conditions must be defined to extract only potential SLCC peaks from the secondary peaks. These conditions were based on the observed statistical distribution of peak properties and were empirically set as: the width of the peak must be < 4 bins (corresponding to 200 m), the peak width height must be > 40 x $10^{-6}$ $m^{-1}$ $sr^{-1}$, and the peak prominence must be > 60 x $10^{-6}$ $m^{-1}$ $sr^{-1}$. The secondary peaks group also include the cases of third and fourth peaks in elevation above the primary peaks when found. Third and fourth peaks were only found in 26 cases (third peak) and 5 cases (fourth peak), representing 0.6% and 0.1% of the cases from all identified peaks. This very small sample size did not allow to use the approach proposed for second peaks and therefore 2nd, 3rd and 4th peaks were all included in the same "secondary peak" group."

Lines 265-270 "The difference between the median value of the single peak distribution and the multiple peak distribution can be calculated and is equal to 4.20 x 10-5 m-1 sr-1 ."
It remains unclear how this was calculated, please clarify. Is there difference between 2nd and 3rd peaks (see previous comment)?

R1C9: This sentence (new lines 303 to 304) now reads as:

"The absolute difference between the median value of the primary peaks distribution ($Q2_{prime}$) and the secondary peaks distribution ($Q2_{second}$) can be calculated as $\left| Q2_{prime} - Q2_{second} \right|$ and is equal to 4.20 x 10-5 $m^{-1}$ $sr^{-1}$ ."

The 25 cases of third peaks, and 5 cases of four peaks both have a wide distribution of peak values, and it is not possible to draw conclusions, given these small sample sizes.

Section 2.3 Enhanced data-driven ceilometer cloud phase mask
More information is needed in this section. Please clarify and describe:
- line 282: what are these "learners"? Please clarify. - How are the training and validation data sets selected? How long training set? How long validation data?
- Can you please tie these ML world terms into observations / data used. Many aspects remains unclear for the people unfamiliar with machine learning and it would be impossible for people to reproduce this algorithm. Any comment on this aspect?

R1C10: Thanks, we are addressing this comment as three points as per below:

(1) "learners": we modified that sentence, and it now reads: "The principle of this algorithm relies on a "boosting" strategy, where predictions of "weak" learners (here, "learners" are decision trees) are combined to produce a "strong" learner by utilising additive training strategies."

(2) "How are the training and validation data sets selected? How long training set? How long validation data?". See also response to R2C19. The split between test and train was implied in the use of the 3 K-fold cross validation approach. This is now explicit, and a new sentence has been added at line 345: "With the 3 k-fold cross validation, two third of the data are allocated to training, while the remaining one third of the data is used for testing."

(3) "Can you please tie these ML world terms into observations / data used. Many aspects remain unclear for the people unfamiliar with machine learning and it would be impossible for people to reproduce this algorithm. Any comment on this aspect?": We added this sentence below in new lines 352-355 to provide information on the implementation of the ML approach. It was already mentioned in the acknowledgements, but we think it is needed here as well, following your comment.

"Data preparation including splitting the data into training and testing for cross validation was implemented using the python library scikit-learn (Pedregosa et al., 2011) and the model XGBoost was implemented using its python library XGBoost (Chen et al., 2015)."

figure 5: would it be possible to show the depolarization values in this plot? Would be interesting to see the data especially in the first half of the data period when T19 method gives lots of SLW detection and reference not.

R1C11: We have now included in Figure 5 a subpanel showing the depolarisation ratio from the Lidar. This is the new subpanel (b) as shown below.

[Figure]

line 435: How good is the reference in the first place? This reference method could be described in more detail as it remain unclear how the reference combination algorithm is working. Is this method seen as reliable reference? Please offer some references.

R1C12: We added a new paragraph to highlight the robustness of our lidar-radar mask at new lines 198 to 205:

"It is necessary to determine the uncertainty in the Raman lidar liquid phase product before quantifying the performance of the ceilometer liquid cloud algorithm. To this end, we performed a Monte Carlo simulation using a random population of N = 1,000 samples from a normally distributed population (Alexander et al., 2021). For each integrated cloud attenuated backscatter $\beta_{int}$ and integrated depolarization ratio $\delta_{int}$ point, we have associated uncertainties $\Delta\beta_{int}$ and $\Delta\delta_{int}$, which we set to be twice the standard deviation of the normal distribution. We then determined the cloud phase for each of these 1,000 realizations. These simulations indicated that we misclassified only around 0.3% of the SLW as ice with the Raman lidar during the three months of observations, showing the robustness of our radar-lidar cloud phase product."

line 538: "raw" means different things for different people. I suggest to clarify, do you mean the data in output files or "attenuated backscatter" - meaning the data after calibration. Is it necessary to calibrate the signal before use?

R1C13: We replaced "raw" by "attenuated". As discussed in the methods section of the paper, we did some calibration as part of the ALCF pipeline, using the attenuated backscatter from the instrument (self-calibration). We do not think it is necessary to go into these details here again in the discussion.

line 539: Just out of curiosity: how many ceilometers there are in Antarctica where no other potential devices are installed?

R1C14: We have sent emails out to the whole of the SCAR Antarctic Clouds and Aerosols Action Group with the aim of collating all the ceilometer data collected around Antarctica. This includes emails to around 25 distinct groups, but the number of sites with only ceilometers deployed will be less than half of this number at present. However, several sites have longer term datasets than those presented in this study so there is some useful spatio-temporal analysis possible potentially.

line 561: "The new ceilometer algorithm described herein has been developed at the Bureau of Meteorology and is not publicly available."
For what use this algorithm is developed if it is not publicly available and cannot be reproduced based on this manuscript?

R1C15: See also R2C27. This was the status at the time of the submission, but our approach has changed since, and we are preparing the algorithms to be included in the ALCF developed by co-author Peter Kuma. It will therefore be available and applicable easily by future users. The text of the manuscript has been changed to:

 "The new G22-Davis ceilometer algorithm described herein as well as the original T1 algorithms are in the process of being included in ALCF and will therefore be open-source and publicly available."

---

## Author Comment (AC2)

amt-2022-10

**" Detection of supercooled liquid water clouds with ceilometers: Development and evaluation of deterministic and data-driven retrievals"**

**Authors' response to comments from Reviewer #2**

**2 May 2022**

We would like to thank the editor and the two reviewers for their very constructive comments on our manuscript. We received genuine insights, which have significantly contributed to increasing the manuscript quality and potential impact. To improve the clarity in our responses we have numbered the reviewers' comments: for example, the comment 1 from reviewer 1 is listed as R1C1 and will refer to these comments as such in the following.

Based on some of the comments from the reviewers and late feedback from our co-authors, we also made small additional changes to the original manuscript:

(1) Following internal discussion between the co-authors, we realised that our initial definition of Supercooled liquid clouds (SLW) was misleading as it included both mixed-phase and SLW clouds. With our technique, and the technique from T19, we can detect clouds containing SLW, e.g., both SLW clouds and mixed phase clouds, as we cannot distinguish between the two. We therefore decided to replace instances where we referred to both SLW and mixed phase as "Supercooled Liquid water Containing Clouds" or SLCC, including in the title.

(2) We have incorporated late feedback from one of our co-authors; these were mostly typo and added precisions in the manuscript and figures.

(3) We have applied our two retrievals to additional data from the same instrument (the ceilometer) that covers a full annual cycle including austral winter months. We have included a new section 3.4 in the manuscript and a new Figure 10. We think that this will add value to the paper, showing how this new retrieval can produce climatology of supercooled liquid water containing clouds in regions such as Antarctica where observations are scarce. We also added a few sentences putting these results into perspective and in the light of other observations in Antarctica and elsewhere.

(4) We made some cosmetic modifications in the Figures: In Figure 1a, the arrow was wrongly labelled "North", we removed the label to Progress 3 and removed the bathymetry. In Figure 5, we changed the x-axis units. Figures 7 and 8 saw the unit of the axis changes to km instead of m to simplify the axis unit labels.

Reviewer #2

The manuscript presents a new machine learning approach for the classification of supercooled liquid water from ceilometer observations. Based on three months observation ins the Arctic, the approach shows improved performance compared to a previous method which also analysis ceilometer profile data. As reference, the authors use a mask derived from a combination of radar and depolarisation lidar observations. The study is nicely presented with convincing scientific quality. To highlight applicability of the novel tool at the large number of ceilometer data being collected globally, the authors could give perspective on the expected performance in other geographical settings and maybe measurements from other ceilometer types. The manuscript can be published after a series of minor comments are addressed.

Line 180: What is the vertical and temporal resolution of the RMAN cloud classification? How is this alight to the 're-gridded' data of Radar and ERA5? How can a classification be 'interpolated' or 'averaged'?

R2C1: The "raw" data from the RMAN lidar is of variable temporal resolution generally ranging from about 30 to 90 seconds, and 15 m vertical resolution. The lidar data are then averaged at 2 min resolution, and the cloud classification is therefore produced at 2 min, 15 m resolution. The cloud radar has a "raw" resolution of 12 s, 25 m. To retain some of the higher cloud radar temporal resolution, the two products are combined to produce a 1 min, 15 m resolution cloud mask, interpolating the radar fields to the vertical resolution of the RMAN and re-sampling the RMAN lidar data to 1 min resolution. The RMAN lidar classification is re-sampled to 1min simply by duplicating the nearest 2min timestep. We added this sentence in the manuscript:

"The radar has a sensitivity of around – 50 dBZ at 1 km, a vertical resolution of 25 m and a sampling frequency of 12 s."

"The "raw" data from the RMAN lidar is of variable temporal resolution generally ranging from about 30 to 90 seconds, and 15 m vertical resolution."

"The original 2 min resolution RMAN lidar classification was re-sampled to 1 min simply by duplicating the nearest 2 min timesteps."

Line 208: Not clear what is meant by 'how to label the 50 m bins.' Please rephrase.

R2C2: We rephrased that sentence, and it now reads: "However, they did not specify how to allocate a classification to the bins at the height of the peak and the surrounding bins (below and above the peak). We decided that based on the above, only the altitude bin corresponding to the location of the peak (if found) was labelled as liquid water."

Line 209: So only one bin is classified as liquid water?

R2C3: This is correct. We did not want to move away from T19, as this serves as our reference mask. In any case, the comparisons are then done timestep to timestep, so the number of bins labelled as SLCC do not interfere in the evaluation.

Line 211: In line 195 you state that the cloud phase mask utilises Ceilometer data only. But now you state that SLW and liquid water is differentiated according to the reanalysis temperature profile. This seems contradictory.

R2C4: The sentence in line 195 has been changed to include discussion of ERA5 and now reads:

"The first cloud phase mask presented herein is based on ceilometer observations, following the work from T19, augmented with ECMWF ERA5 interpolated temperature fields to differentiate SLW from other liquid water."

Line 239: if you say minimum peak width is 50m , this means only one range bin as you are operating an a grid of 50m?

R2C5: We have changed the sentence to: "….least a width of 50 m (thus only one range bin since this is the lowest resolution of our post-processed ceilometer data), and a peak…"

Line 241: Where do you define the peak width? At half maximum or base?

R2C6: We added the equation that explained how the peak width height is calculated to the text:

"The height at which the peak width is measured is relative to its prominence, following eq. (1):

$$H_{peak\ width} = H_{height} - Prom \times RH \tag{1}$$

With $H_{peak\ width}$ the height of the peak at which the width is measured (m$^{-1}$ sr$^{-1}$), $H_{peak}$ the absolute height of the peak, $Prom$ is the prominence (m$^{-1}$ sr$^{-1}$),  and $RH$ the relative height, which was set at 0.5."

Line 244: 'Lowest' peak defined by peak magnitude or altitude?

R2C7: it now reads:" …with the lowest peak in altitude taking the number '0'".

Line 245: Again, you are using ERA5 temperatures. I think you need to be careful calling the algorithm to utilise "ceilometer data only"?

R2C8: This has been changed in R2C4 and elsewhere in the manuscript (abstract and introduction) to include mention of ERA5 temperature.

Line 254: rephrase "For single peaks, SLW data-only were selected based on the Boolean condition defined using the radar-lidar cloud mask". What is meant by "data-only"?

R2C9: This has been re-phrased as:

"For single peaks, data for which SLCC were identified were selected based on the Boolean condition defined using the radar-lidar cloud mask."

Line 255: remove "arbitrary"? Your conditions have an empirical basis.

R2C10: Indeed! We replaced "arbitrary" by "empirically based".

Line 257: what is meant by "width < 4"? Bins? Maybe better to use width in units of meters?

R2C11: It now reads: "the width of the peak must be < 4 bins (200 m)".

Line 265: Why is the multiple-peak distribution so narrow? Should there not be a dependence on the order of peak in the profile? i.e. could be not expect the peak at the lowest altitude in a multiple-peak profile to resemble the signature of a single-peak? Do you account for the order or altitude of the peaks?

R2C12: Apologies, the terminology "multiple peaks" has been mis-leading. Please see response to R1C8 and R1C9 where we address these issues.

Line 274: how often do you find this mismatch between peak-criteria and cloud classification mask that leads to an adjustment of the "true" indicator? What does this mean physically?

R2C13: Thanks to our clarification regarding the terminology used for "multiple peaks" (responses R1C8 and R1C9), our statement here line 279 shall be seen with that clarified definition in mind. This case is when typically for a given timestep when the primary peak is identified as SLCC, and the secondary peak(s) is(are) not. The label for the secondary peak(s) should then be revised as non SLCC.

Caption Figure 4: Introduce meaning of 'ts'.

R2C14: Done.

Line 330: This seems like an artificial problem. The masks are created based on higher-resolution data. Why would you create a 50m vertical resolution grid for the ceilometer-based mask if the observations have a resolution of 10m? would it not be more appropriate to map all data to the same vertical resolution in the beginning so they could now be compared more easily?

R2C15: The resolution of 5 min, 5 0m was found optimal in ALCF to reduce noise in the attenuated backscatter. Using a higher temporal and spatial resolution will increase the noise to signal ratio. Therefore, we decided to use that same re-sampling strategy. For the radar-lidar mask, the 15 m bin resolution followed the same approach as in Alexander and Protat (2020). Bin to bin comparisons will be affected by many factors and is not the objective of our work here.

Line 334: same question for the temporal resolution.

R2C16: See also R1C3. We added this sentence to the manuscript:

"Subsampling is mostly done to improve signal-to-noise ratio. The cloud masking usually benefits from subsampling to 5 min intervals and 50 m vertical resolution, because it decreases the number of misclassified bins;"

Line 352: introduce meaning of confusion mask indicators

R2C17: We have inverted the order of the definition in that section (2.5) and have introduced the confusion matrix indicators at the end, after the basic definitions have been provided.

Line 357: state clearly what you are referring to with the term "prediction". Is true negative the case when the mask correctly indicates the absence of SLW? Then why call this "wrong prediction"? Also, if "false positive" refers to the mask wrongly assigning SLW, then why would you call this "wrongly indicating a correct prediction". Please clarify this paragraph.

R2C18: Thanks, indeed that wasn't very clear… we have modified that paragraph and it now reads:

"A true positive (TP) is defined as a test result indicating a correct prediction (correctly predicting the occurrence of SLCC), a true negative (TN) is defined as a test result correctly predicting the absence of SLCC, while a false positive (FP) is defined as a test result wrongly predicting the presence of SLCC, and a false negative (FN) is a test result wrongly indicating the absence of SLCC."

Line 388: what is the number of samples in the training data? Is this stated in the methods section?

R2C19: It was implied in the method, as 3 k-fold cross validation by definition allocates two third of the total data for training and one third for testing. We have added a sentence in the methodological section (new lines 318 319) to explicitly state this:

"With the 3 k-fold cross validation, two third of the data are allocated to training, while the remaining one third of the data is used for testing."

Line 411: Of course figures should be explained when they are being discussed, but please avoid repeating content of figure captions in the text.

R2C20: This sentence now reads: "In Figure 7, the average peak properties for which peaks had been identified for the 6th of January 2019 are presented as joint distributions using kernel density estimation plots with peak β value as the y-axis". The introduction of Figure 8 in the text was also changed similarly. We also simplified the text for the introduction of Figure 9 to avoid repetition between figure captions and the text.

Line 424: How did you evaluate presence of fog?

R2C21: We refer here to the fog identified based on our algorithm as described in the methodological section, such as: "The same method as T19 was again used here, detecting fog layers by identifying values of backscatter above $ß = 10^{-5}$ $m^{-1}$ $sr^{-1}$ for the lowest grid point (corresponding to 0-50 m above the surface) and a ß value 250 m above the instrument of $ß < 3 \times 10^{-7}$ $m^{-1}$ $sr^{-1}$ (to restrict the identification to fog, and exclude low-level thicker clouds)." We did not refer to independent observations to confirm or not its presence.

Line 444: Given you are using various different products, please use consistent labels throughout. E.g. in a similar way you are using T19, please use one label (such as XGBoost) for the "new algorithm".

Also, please use one consistent label for the reference data. Right now, the reader can get easily confused. E.g. here I am wondering if "a data-driven threshold approach" has already been introduced or if this is yet another method.

R2C22: We have decided to label our new approach "G22-Davis" and refer as such in the manuscript. This terminology is now introduced in line 333:

"To facilitate the discussion in the next sections of this study, we further refer to our algorithm as G22-Davis. The extension "-Davis" illustrates that our G22 model had been trained with data collected at Davis, and we can imagine that the same model could be applied to data collected elsewhere."

Subsequently, there are 12 instances where we have replaced the unclear "new algorithm" terminology by "G22-Davis".

Line 473: So the thresholds are not actually "arbitrary", but rather empirical values determined based on the previous analysis. The fact that they work well for your dataset is hence not surprising. It would now be the next step to assess whether these thresholds are more widely applicable, e.g. to perform SLW detection for a different time period or different location.

R2C23: Yes, that's correct. We replaced "arbitrary" by "empirical". Following your comment, we have added a sentence in the discussion section:

"Finally, in the absence of radar-lidar mask to train and test a model like in G22-Davis, there remains the possibility of using the classification approach based on thresholds derived from peak characteristics joint distributions. This method should be assessed for different periods at Davis or for other locations than Davis, to test of these threshold values are widely applicable."

Line 484: After presenting these values, please put results into context e.g. to performance of the other approaches.

R2C24: This now reads: "This is an improvement of 0.07 as compared to the accuracy of T19, which was equal to 0.84. For the dataset for which peaks were identified, the total dataset was made of 11,327 datapoints. The best testing score was of 0.81 (with learning rate = 0.01, max depth = 12, child weight = 8), for training accuracy scores (or f1) of 0.94. This is a substantial improvement of almost 0.1 as compared to the accuracy of T19, which was equal to 0.72."

Line 507: This seems to contradict your statement from line 453: "The value of $\beta$ at peak is directly correlated to the peak width height, making that feature redundant." Please explain.

R2C25: in former line 453, we changed the text to: "The value of $\beta$ at peak versus peak prominence is not presented as these are directly correlated. The value of $\beta$ at peak is correlated to the peak width height, with some differences."

The redundant feature is peak prominence, rather than peak width height, as can be seen in Figure 9.

Line 509: If peak temperature is not an important predictor, would it be possible to omit the ERA5 data and work solely on ceilometer observations as input?

R2C26: We added that sentence in new lines 594 to 597:

"Given the low importance of air temperature for accurate prediction of SLCC, we could consider not using that feature as an input to G22-Davis and removing our dependency on ERA5 or other NWP inputs. However, at Davis, we might likely be in the specific case where air temperatures are often too negative to produce liquid water droplets (other than supercooled) as seen in the T19 cloud phase classification. For other climates with higher air temperatures, the air temperature feature might be more relevant."

Line 562: You are using software and algorithms developed elsewhere, yet you are not intending to share the code? Especially as you are claiming your algorithm has better performance than an existing approach of T19, it would be important to the community to be able to test your algorithm and verify your findings.

R2C27: See also R1C15. This was the status at the time of the submission, but our approach has changed since, and we are preparing the algorithms to be included in the ALCF developed by co-author Peter Kuma. It will therefore be available and applicable easily by future users. The text of the manuscript has been changed to:

"The new G22-Davis ceilometer algorithm described herein as well as the original T1 algorithms are in the process of being included in ALCF and will therefore be open-source and publicly available."

Figure 6: how is the "baseline" determined based on which you quantify the "peak prominence"?

R2C28: We added these two sentences in the figure caption:

"The baselines (lowest points on the green lines) were calculated as the lowest contour lines around the peak. To identify the peak characteristics, we used the signal processing tools of the python library SciPy."

Figure 7 and Figure 8: these are not a "scatterplots" because the individual sample pairs are not shown. Rather you are comparing isolines for the two cases.

R2C29: See R2C20. We also changed the captions of Figure 7 and Figure 8 accordingly and refer to the plots as: "Joint distribution using kernel density estimation plots".